# Defining a conformational ensemble that directs activation of PPARγ

Ian M. Chrisman[1,2], Michelle D. Nemetchek[1,2,3], Ian Mitchelle S. de Vera[4,8], Jinsai Shang[4], Zahra Heidari[2,3], Yanan Long [4,5], Hermes Reyes-Caballero[4], Rodrigo Galindo-Murillo [6], Thomas E. Cheatham, III[6], Anne-Laure Blayo[7], Youseung Shin[7], Jakob Fuhrmann[4], Patrick R. Griffin[4,7], Theodore M. Kamenecka[7], Douglas J. Kojetin [4,7] & Travis S. Hughes [2,3]

The nuclear receptor ligand-binding domain (LBD) is a highly dynamic entity. Crystal structures have defined multiple low-energy LBD structural conformations of the activation function-2 (AF-2) co-regulator-binding surface, yet it remains unclear how ligand binding influences the number and population of conformations within the AF-2 structural ensemble. Here, we present a nuclear receptor co-regulator-binding surface structural ensemble in solution, viewed through the lens of fluorine-19 ($^{19}$F) nuclear magnetic resonance (NMR) and molecular simulations, and the response of this ensemble to ligands, co-regulator peptides and heterodimerization. We correlate the composition of this ensemble with function in peroxisome proliferator-activated receptor-γ (PPARγ) utilizing ligands of diverse efficacy in co-regulator recruitment. While the co-regulator surface of apo PPARγ and partial-agonist-bound PPARγ is characterized by multiple thermodynamically accessible conformations, the full and inverse-agonist-bound PPARγ co-regulator surface is restricted to a few conformations which favor coactivator or corepressor binding, respectively.

[1] Biochemistry and Biophysics Graduate Program, The University of Montana, Missoula, MT 59812, USA. [2] Center for Biomolecular Structure and Dynamics, The University of Montana, Missoula, MT 59812, USA. [3] Department of Biomedical and Pharmaceutical Sciences, The University of Montana, Missoula, MT 59812, USA. [4] Department of Integrative Structural and Computational Biology, Scripps Florida, The Scripps Research Institute, Jupiter, FL 33458, USA. [5] Summer Undergraduate Research Fellows (SURF) Program, Scripps Florida, The Scripps Research Institute, Jupiter, FL 33458, USA. [6] Department of Medicinal Chemistry, University of Utah, Salt Lake City, UT 84112, USA. [7] Department of Molecular Medicine, Scripps Florida, The Scripps Research Institute, Jupiter, FL 33458, USA. [8] Present address: Department of Pharmacology & Physiology, Saint Louis University School of Medicine, Saint Louis, MO 63104, USA. These authors contributed equally: Michelle D. Nemetchek, Ian Mitchelle S. de Vera, Jinsai Shang. Correspondence and requests for materials should be addressed to T.S.H. (email: travis.hughes@umontana.edu)

Nuclear receptors are ligand-regulated transcription factors that mediate the transcriptional actions of lipophilic endogenous ligands, including steroid hormones and lipids[1], and are the target of ~13% of US Food and Drug Administration (FDA)-approved drugs[2]. The binding of these natural ligands, as well as synthetic ligands and FDA-approved drugs, to the nuclear receptor ligand-binding domain (LBD) affects the recruitment of transcriptional co-regulator proteins to target gene promoters, which influences chromatin remodeling and gene transcription[3]. Crystal structures of nuclear receptor LBDs have revealed in exquisite detail the molecular contacts created between the receptor and ligand, as well as low-energy active and inactive conformations of helix 12[4–7]. Helix 12 is a critical regulatory structural element in the activation function-2 (AF-2) co-regulator interaction surface of many nuclear receptors[8]. Over 100 crystal structures have been solved of the peroxisome proliferator-activated receptor-γ (PPARγ) LBD bound to ligands of various scaffolds and pharmacological activities[9]. Surprisingly, the backbone conformations of these structures, in particular the conformation of helix 12, are all very similar despite the fact that PPARγ is bound to ligands that produce a diverse range of functional outputs. Thus, it is difficult to understand the structural mechanism of action by which the binding of ligands with diverse activities affect helix 12 conformation from crystallography data alone. One hypothesis is that helix 12 consists of a dynamic ensemble of conformations, and not one single or static conformation in the presence or absence of a bound ligand[10, 11]. However, experimental evidence describing this ensemble is lacking and it remains poorly understood how binding of pharmacologically distinct ligands affects the ensemble of co-regulator-binding surface and helix 12 conformations.

Solution structural methods indicate prevalent, ligand-dependent helix 12 movement. Hydrogen deuterium exchange mass spectrometry (HDX-MS) demonstrates a relationship between helix 12 stability and agonist binding for nuclear receptors[12–15]. Nuclear magnetic resonance (NMR) studies implicate movement on the microsecond–millisecond (μs–ms) time scale between two or more conformations over a large portion of the apo PPARγ LBD and partial-agonist-bound PPARγ LBD. These movements result in very broad or unobserved NMR resonances that prohibit structural analyses. Full agonists robustly diminish these dynamics[16–19]. Furthermore, crystal structures, HDX-MS, and protein NMR have provided complementary information revealing a relationship between structure and function for PPARγ (e.g., the presence or absence of critical hydrogen bonds between ligand and helix 12[20]); however, a direct observation of the ligand-dependent ensemble implied by these data is lacking. This raises the question: are there multiple long-lived conformations that correlate with functional efficacy (e.g., co-regulator affinity) in nuclear receptors?

It remains challenging to quantify the number, relative population, and kinetics of exchange between the conformations that compose this putative ensemble and how the ensemble is influenced by binding small molecules and co-regulators. $^{19}$F (fluorine-19) NMR spectroscopy is exceptionally sensitive to structural and environmental changes, can reveal structural information from regions of a protein that are unobserved via 2D/3D NMR[21], and can be used to probe how ligands affect the conformational ensemble of proteins[22–26]. Here, using $^{19}$F NMR combined with biochemical co-regulator interaction analysis and molecular simulations, we define the ligand-dependent conformational ensemble of the co-regulator interaction surface, including helix 12, which controls the transcriptional activity of PPARγ. The data presented here indicate that helix 12 and the co-regulator-binding surface of apo PPARγ and partial-agonist-bound PPARγ is found in a broad energy well with multiple local minima of similar potential energy separated by relatively small kinetic barriers, allowing exchange on the μs to ms time scale. In contrast, when PPARγ is bound to a full agonist or inverse agonist, helix 12 and the co-regulator-binding surface occupies narrow energy wells with fewer thermodynamically accessible conformations. In addition, simulations define some of the probable structures that compose these ensembles. These data better elucidate how ligands induce functional effects via nuclear receptors.

## Results

**Diverse activities of synthetic PPARγ ligands.** We assembled a set of 16 pharmacologically distinct PPARγ ligands that we and others developed or previously characterized in cellular and structure–function studies (Supplementary Fig. 16) including full and partial agonists that robustly or mildly enhance transcriptional activation; antagonists/non-agonists that block activation or maintain constitutive basal cellular transcriptional activity; and inverse agonists that repress transcription compared to the basal receptor activity. In a time-resolved fluorescence resonance energy transfer (TR-FRET) co-regulator interaction assay, full agonists such as GW1929 and rosiglitazone that induce robust PPARγ transcription[27, 28] increase binding of a peptide derived from mediator of RNA polymerase II transcription subunit 1 (MED1) coactivator (Fig. 1a), and decrease binding of a peptide derived from the nuclear receptor corepressor 1 (NCoR1) referred to herein as NCoR (Fig. 1b). In contrast, ligands that function as inverse agonists such as T0070907 and SR10221 lower PPARγ transcriptional responses relative to basal cellular activity[13, 29, 30], increase binding of NCoR (Fig. 1b), and decrease binding of MED1 (Fig. 1a). Crystal structures of ligand-bound PPARγ LBD typically show a non-crystallographic dimer configuration containing two chains, A and B (Supplementary Fig. 2a). The main difference between the two chain A and chain B protein molecules involves helix 12, which adopts distinct conformations commonly referred to as active and inactive. The active conformation is assumed to be the conformation in solution when bound to a full agonist that induces increased transcription. Notably, these active and inactive helix 12 conformations are both influenced by crystal contacts (Supplementary Fig. 2b).

**NMR-detected co-regulator-binding surface structural ensemble.** To facilitate $^{19}$F NMR studies of the co-regulator-binding surface, which is composed of portions of helix 3, 4, and 12, we introduced a cysteine residue on the C terminus of helix 12 at several locations (K502C, Y505C, and Q498C) and on helix 3 (Q322C) of the PPARγ LBD to allow covalent linkage of 3-bromo-1,1,1-trifluoroacetone (BTFA), a small molecule containing a trifluoromethyl ($-CF_3$) group. Q498C caused protein instability, whereas BTFA attached to Y505C did not show pronounced ligand-induced changes to the $^{19}$F NMR spectra of the PPARγ LBD, presumably due to its position at the unstructured C terminus of helix 12 (Supplementary Fig. 1). However, Q322C and K502C yielded well-functioning protein with pronounced ligand-inducible changes; we used these mutants to probe the conformational ensemble of the PPARγ co-regulator-binding surface (i.e., the AF-2 surface).

Molecular simulations indicate that K502 (wild-type (wt) residue) and K502C-BTFA (modified residue) are both solvent exposed in the active helix 12 conformation and are not likely to sterically hinder co-regulator binding in this active conformation (Fig. 2a, Supplementary Fig. 3). Control experiments indicate that the introduced cysteine on helix 12 (C502) is preferentially labeled over the only native cysteine (C313), which could be because C313 points into the ligand-binding pocket

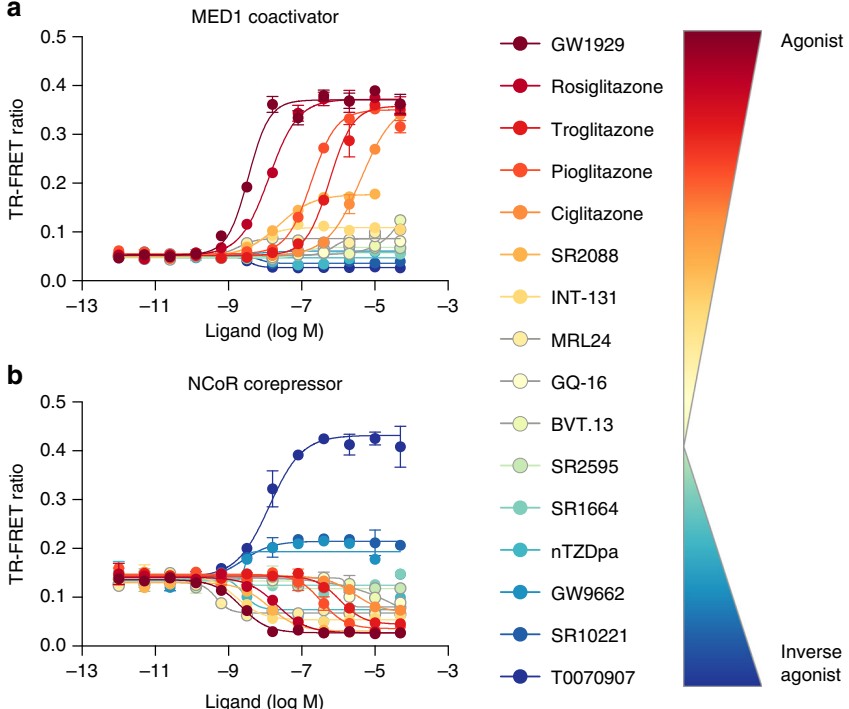

**Fig. 1** Nuclear receptor co-regulator interaction differentiates pharmacologically distinct PPARγ ligands. TR-FRET biochemical assay shows the effect of the compounds on the interaction between PPARγ LBD and peptides derived from the **a** MED1 coactivator and **b** NCoR corepressor, plotted as TR-FRET ratio (665 nm/620 nm) vs. ligand concentration ($n = 2$, standard deviation). The data shown represents technical replicates from a single experiment and the experiment was repeated four times with similar results. The window of efficacy in these data is representative of ligand-induced changes in co-regulator affinity for PPARγ. An increase in TR-FRET ratio indicates a strengthening of affinity for MED1/NCoR compared to apo while a decrease indicates a ligand-induced weakening of affinity for the co-regulator. The effect of vehicle (DMSO) is negligible (Supplementary Fig. 5); furthermore, the DMSO concentration is constant across the titration both in this figure and all other TR-FRET data presented

(Supplementary Fig. 4). We refer hereafter to BTFA-labeled PPARγ K502C protein as PPARγ$^{K502C}$-BTFA, we also use PPARγ$^{K502C}$-BTFA with a C313A mutation (referred to as PPARγ$^{C313A,K502C}$-BTFA) for comparison, as this protein can only be labeled on helix 12. $^{19}$F NMR signals in these labeled proteins could conceivably arise from BTFA-labeled co-purified protein impurities; however, the spectrum of PPARγC313A, which lacks all cysteines, reveals no detectable signal from impurities (Supplementary Fig. 5).

Next, we determined the effects of mutations and labeling on the function of the PPARγ LBD. First, we measured ligand $K_i$ values using a competitive ligand displacement assay to determine if the mutations and BTFA label affects ligand affinity. Compared to wt, the PPARγ$^{K502C}$-BTFA and PPARγ$^{C313A,K502C}$-BTFA proteins exhibit a 5-fold and 11-fold median reduction in ligand affinity, respectively (Supplementary Table 1); however, only four ligands (GQ-16, BVT.13, ciglitazone, and troglitazone) are predicted to be at <93% occupancy in our samples under the conditions used for the NMR experiments reported here (Supplementary Table 2). Second, we measured co-regulator recruitment efficacy and half-maximal effective concentration (EC$_{50}$) using TR-FRET. As expected from the calculated $K_i$ values, PPARγ$^{K502C}$-BTFA has a 2-fold and 5-fold median reduction in EC$_{50}$ and PPARγ$^{C313A,K502C}$-BTFA has a 1-fold and 9-fold median reduction in EC$_{50}$ in recruiting MED1 and NCoR, respectively, compared to wt PPARγ (Supplementary Table 3). Importantly, ligand-induced co-regulator recruitment efficacy is highly correlated for PPARγ$^{K502C}$-BTFA and wt, indicating that PPARγ$^{K502C}$-BTFA is functionally similar to wt PPARγ LBD ($R^2$ = 0.8 for NCoR and $R^2$ = 0.98 for MED1; Supplementary Fig. 6d, f). In contrast, the relative NCoR recruitment efficacy differs

between PPARγ$^{C313A,K502C}$-BTFA and wt, while the relative MED1 recruitment efficacy is comparable to wt in this labeled mutant ($R^2$ = 0.11 for NCoR and $R^2$ = 0.8 for MED1; Supplementary Fig. 6d, f). We also found that the mutations and labeling had little effect on recruitment efficacy and EC$_{50}$ of a coactivator peptide derived from CREB-binding protein (CBP) for both labeled forms of PPARγ (Supplementary Fig. 6b, e and Supplementary Table 4).

In addition, we compared mutant and wt PPARγ affinity for fluorescein isothiocyanate (FITC)-labeled NCoR, MED1, and a peptide from the silencing mediator of retinoic acid and thyroid hormone receptor (SMRT) corepressor utilizing fluorescence polarization (FP). Labeling did not significantly affect affinity of NCoR or MED1 for the four tested PPARγ–ligand complexes (Supplementary Fig. 7), and consistent with the TR-FRET data, PPARγ$^{K502C}$-BTFA affinity for NCoR and MED1 correlated most closely with wt PPARγ. Both labeled PPARγs affect SMRT peptide affinity, although PPARγ$^{K502C}$-BTFA does so to a lesser extent, indicating that the label may directly interfere with SMRT binding (Supplementary Fig. 7). In general, these data indicate that the relative co-regulator affinities are consistent between wt and especially PPARγ$^{K502C}$-BTFA. Given these data, we focused on PPARγ$^{K502C}$-BTFA to correlate structure with function and utilize PPARγ$^{C313A,K502C}$-BTFA to confirm specific labeling of PPARγ$^{K502C}$-BTFA and in cases where increased signal to noise is required. PPARγ$^{K502C}$-BTFA is incompletely labeled (Supplementary Fig. 4d) to avoid labeling C313 (Supplementary Fig. 4a–c), which decreases the NMR signal. In addition, PPARγK502C that is not labeled with BTFA is not TR-FRET active (Supplementary Fig. 4e), but has the same affinity for co-regulators as PPARγ$^{K502C}$-BTFA (Supplementary Fig. 7b);

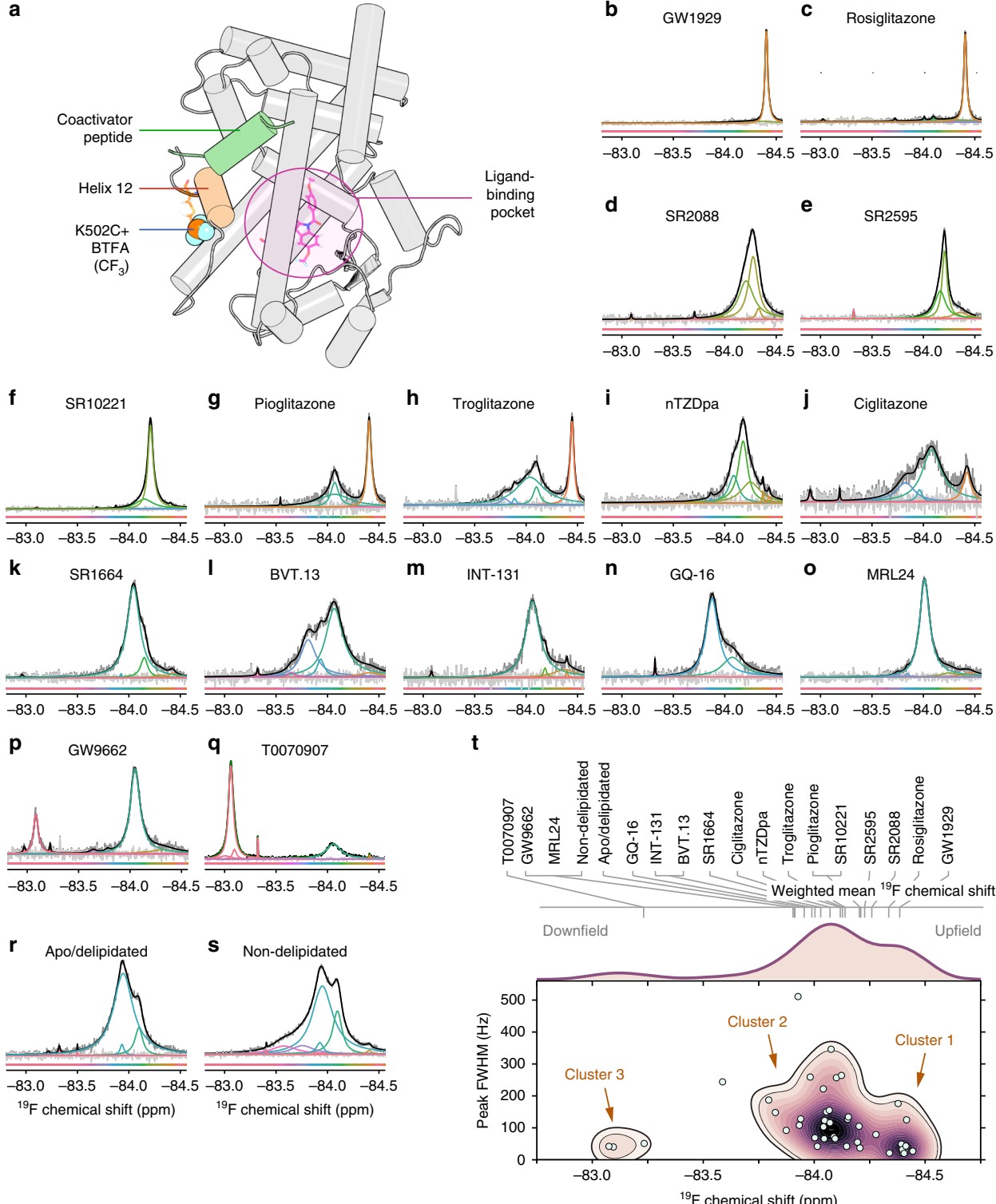

**Fig. 2** [19]F NMR analysis of PPARγ[K502C]-BTFA bound to 16 pharmacologically distinct ligands. **a** Location of covalently attached BTFA tag. **b–s** [19]F NMR spectra (medium gray lines) of PPARγ[K502C]-BTFA **b–q** bound to ligand and ordered according to mean-weighted [19]F chemical shift or **r** delipidated and **s** non-delipidated apoprotein. Fitted peaks are colored according to fitted [19]F chemical shifts; deconvoluted spectra and residuals are shown in black and light gray lines, respectively. Some of these spectra have been replicated (Supplementary Fig. 10). **t** Plot of the full width half max (FWHM) of the major fitted peaks (only peaks that comprise >5% of the spectrum area are included) vs. fitted chemical shifts. Clusters were detected by bivariate kernel density estimation (purple contours) with mean-weighted [19]F chemical shift values annotated on top of the plot. Computational simulations were run of labeled PPARγ (PPARγ[K502C]-BTFA) bound to one of these 16 ligands (GW1929) that demonstrate that the tag stays on or near the surface of the protein (Supplementary Fig. 3)

therefore, any unlabeled portion of PPARγ[K502C]-BTFA should not affect FP or TR-FRET results.

We collected [19]F NMR spectra of PPARγ[K502C]-BTFA bound individually to each of the 16 synthetic ligands (Fig. 2b–q), as well as spectra of apoprotein with and without delipidation (Fig. 2r, s). Bacterially expressed PPARγ often contains fatty acids, which can have functional effects[31], and therefore we remove these lipids; additional data regarding the effects of delipidation on structure and function are presented below. [19]F NMR provides a time-averaged view of the conformational ensemble of helix 12, thus long-lived major conformations (>ms lifetime) show up as distinct peaks, while conformations with lifetimes on the µs–ms time scale show up as a broad single peak. A single conformation (i.e., fast exchange among minor conformational variants) produces a single narrow peak. We used an objective deconvolution[32] method to determine the number of peaks and corresponding peak line widths that compose the recorded spectra. The line widths of the peaks obtained from the objective deconvolution are consistent with measured NMR lifetimes of the [19]F ($T_2$) that we obtained for several liganded states (Supplementary Table 5). Bivariate kernel density estimation of all the deconvoluted spectra revealed two primary clusters of peaks, or conformations and a third lowly populated cluster (Fig. 2t). In general, ligands known to be efficacious or strong agonists (e.g., GW1929, rosiglitazone, pioglitazone, and troglitazone) populate the most upfield (right) group of the narrowest peaks in cluster 1. Ligands of other pharmacological types, including partial agonists and inverse agonists, are found in cluster 2, which is composed of wider peaks, indicative of ensembles composed of multiple conformations. A third lowly populated cluster of downfield chemical shifts and relatively narrow peak widths, cluster 3, occurs for T0070907 and GW9662 ligands that covalently bind to

C313 in the canonical ligand-binding pocket and function as inverse agonists in cell-based assays[29, 30]. These covalent ligands do not attach to the introduced cysteine on helix 12 (C502; Supplementary Fig. 5a, b). PPARγ[C313A,K502C]-BTFA loaded with or without the same 16 ligands show very similar [19]F NMR spectra to PPARγ[K502C]-BTFA, except for ligands which covalently bind to C313, which as expected look similar to ligand-free protein/apoprotein because PPARγ[C313A,K502C]-BTFA lacks C313 (Supplementary Fig. 8). Overall, these data indicate a diverse ligand-dependent helix 12 conformational ensemble.

A contribution to the ligand-dependent differences in [19]F NMR chemical shift values could be the relative solvent exposure of the BTFA probe –CF₃ group, which can be determined by collecting [19]F NMR spectra as a function of deuterium oxide ($D_2O$) concentration. When increasing amounts of $D_2O$ are added to a sample, a large upfield shift (i.e., to the right) of the [19]F NMR peak indicates high solvent exposure of the BTFA probe, while a smaller shift indicates low solvent exposure[26], for example, a solvent-exposed 5-fluorotryptophan shifted 0.217 ppm upfield in 90% $D_2O$ buffer[33]. The chemical shift of free BTFA, which has high solvent exposure, has nearly the largest chemical shift change of any measured here (0.115 ppm upfield), whereas the solvent-protected MRL24 ligand CF₃ group, which is buried deep in the ligand-binding pocket, moves in the opposite direction with increasing $D_2O$ concentrations (Fig. 3 and Supplementary Fig. 13). Given the solvent-protected position of MRL24 CF₃ in the ligand-binding pocket[34], the movement in the opposite direction of the MRL24 ligand CF₃ group likely indicates interaction between the ligand CF₃ group and the protein that is perturbed upon addition of $D_2O$[35]. GW1929-bound PPARγ[C313A,K502C]-BTFA has the largest upfield chemical shift (0.116 ppm; Fig. 3), which is consistent with molecular simulations that show

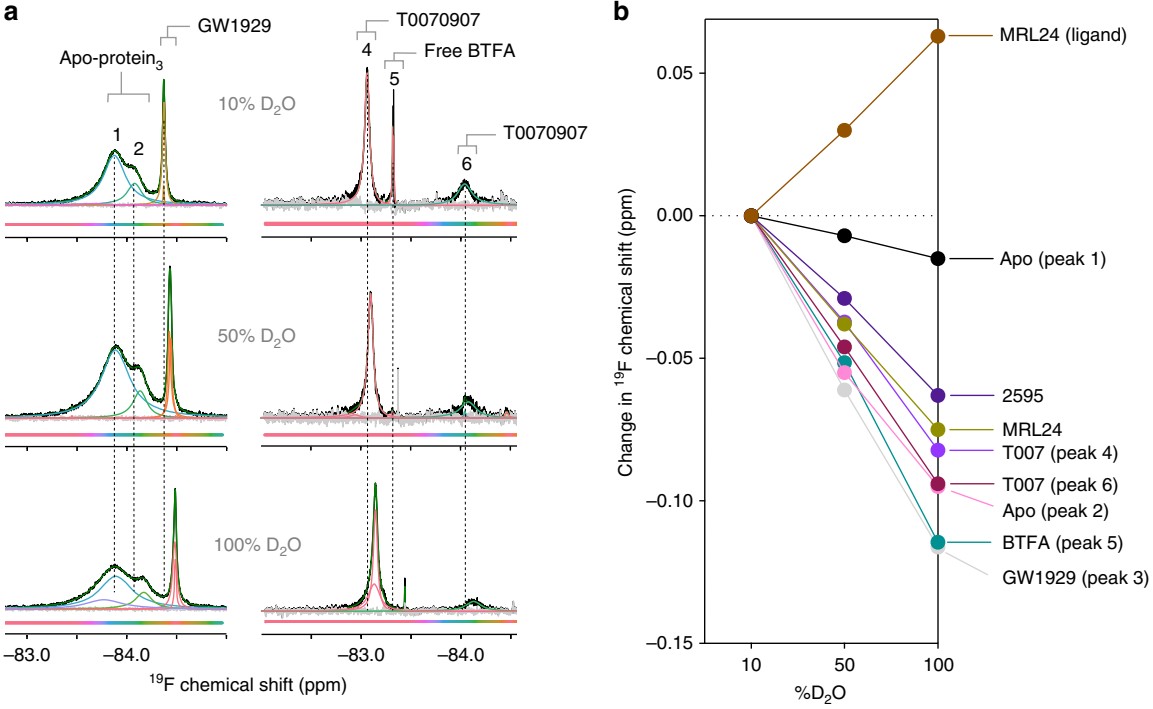

**Fig. 3** Helix 12 solvent exposure is distinct for the active and inactive helix 12 conformations. **a** [19]F NMR spectra of delipidated apo PPARγ[C313A,K502C]-BTFA bound to 0.2 molar equivalent of GW1929 (left), and T0070907-bound PPARγ[K502C]-BTFA with excess free BTFA, at the indicated concentrations of $D_2O$. Due to the high binding affinity of GW1929 (4 nM), all three expected peaks (two apoprotein peaks and one GW1929 peak) are present in slow exchange on the NMR time scale, allowing analysis of the three conformations simultaneously. **b** $D_2O$ solvent isotope-induced changes, plotted as % $D_2O$ vs. change [19]F NMR chemical shift values for various peaks; a dotted gray line is shown to highlight no $D_2O$-induced change in [19]F NMR chemical shift. These experiments were performed once

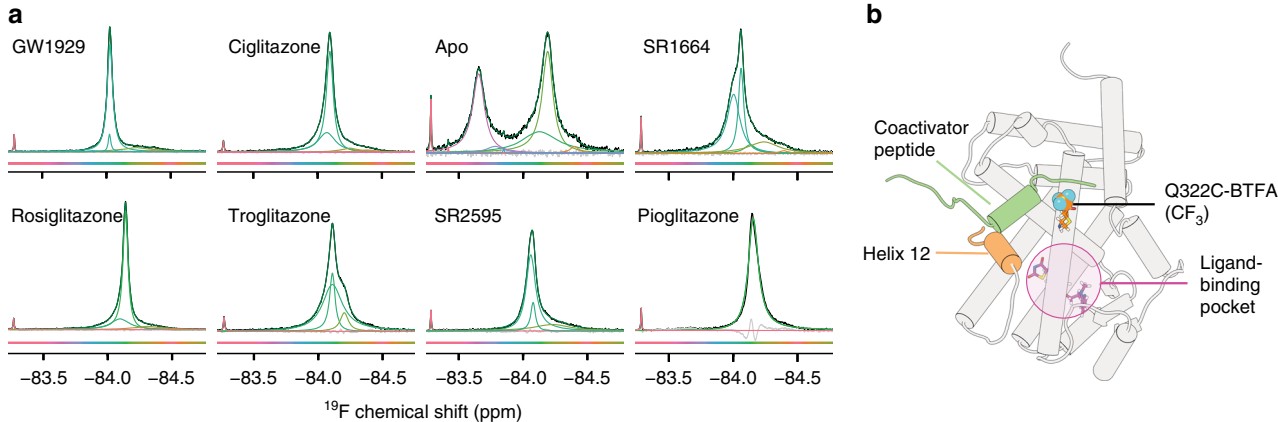

**Fig. 4** Agonists reduce the conformational complexity of the co-regulator-binding surface. **a** Fluorine NMR spectra of PPARγ$^{C313A,Q322C}$-BTFA bound to the indicated ligands. The small sharp left-shifted peak in all the spectra is free BTFA. **b** Trajectory frame from a simulation of PPARγ$^{C313A,Q322C}$-BTFA bound to a coactivator peptide (MED1; green) with 322C-BTFA shown in orange as spheres (fluorine atoms are turquoise). These experiments were performed once

that K502 and C502 both have considerable solvent exposure in the active conformation (Supplementary Fig. 3). These data indicate that full agonists primarily populate the active crystalized conformation in solution, while apo, partial agonists, and inverse agonists are found in other distinct conformations.

PPARγ protein expressed in bacteria can pull down medium-chain fatty acids, which function as weak partial agonists[31]. In agreement, non-delipidated apoprotein that is partially bound to endogenous *Escherichia coli* lipids (Fig. 2s) afforded $^{19}$F NMR spectra with two peaks of similar chemical shift to delipidated apoprotein (Fig. 2r) along with several other lowly populated peaks. Comparison of 2D NMR spectra of delipidated and non-delipidated PPARγ LBD and PPARγ$^{C313A,K502C}$-BTFA in apo form or bound to ligands indicate that *E. coli* lipids have relatively minor effects on helix 12 and backbone structure (Supplementary Fig. 9a, b). In addition, TR-FRET data collected using non-delipidated or delipidated PPARγ-LBD shows essentially the same ligand-dependent co-regulator recruitment potency and efficacy (Supplementary Fig. 9c, d). In separate protein preparations we did observe some variation in T0070907-bound spectra (Supplementary Fig. 10c), which could be due to variable amounts of residual co-bound lipids remaining after delipidation as PPARγ has a very large ligand-binding pocket that can accommodate more than one bound ligand[18]. Overall, these data indicate that some inverse agonists do more than simply displace activating lipids, but instead induce a distinct PPARγ LBD state with higher affinity for corepressors than delipidated apoprotein.

We probed ligand-induced changes in another region of the PPARγ co-regulator-binding surface using a Q322C mutation, which is located in helix 3, generating PPARγ$^{C313A,Q322C}$-BTFA. The fluorine probe in this variant is solvent exposed in the active conformation and maintains wt-like recruitment of co-regulators (Fig. 4 and Supplementary Fig. 11). The chemical shift difference induced by ligands is relatively small; however, just as with the helix 12 probe, ligands decrease the conformational diversity of this region with the strongest agonists (rosiglitazone and GW1929), yielding the narrowest peaks indicative of a single main conformational ensemble. Similar to apo PPARγ$^{K502C}$-BTFA (Fig. 2r), apo PPARγ$^{C313A,Q322C}$-BTFA produces a spectrum with two main wide peaks, indicative of two sub-ensembles composed of multiple conformations with slow exchange between the sub-ensembles (Fig. 4). These data indicate that the ligand-free co-regulator-binding surface is composed of two main structurally diverse (i.e., wide NMR peak) ensembles in slow exchange. Alternatively, one of the peaks could be PPARγ

bound to residual *E. coli* lipids; however, this is unlikely given that the apo spectrum is stable over time (Supplementary Fig. 11), whereas *E. coli* lipid-bound PPARγ$^{K502C}$-BTFA spectra change with time (Supplementary Fig. 10a). The exchange rate between these two apo NMR peaks is discussed below.

**Slow exchange between ensemble structures.** A single well-populated $^{19}$F NMR peak is observed for BTFA probes placed in two areas of the co-regulator-binding surface (helices 3 and 12) when PPARγ-BTFA is bound to a strong agonist such as GW1929 or rosiglitazone. However, for less efficacious ligands, multiple-well-populated $^{19}$F NMR peaks are observed in slow exchange on the NMR time scale indicating that the peaks represent distinct co-regulator-binding surface conformations with lifetimes on the order of milliseconds or longer.

To confirm the apparent slow exchange between different conformations, we performed $^{19}$F chemical exchange saturation transfer NMR experiments on PPARγ where multiple peaks, or conformations, are present (Fig. 5a and Supplementary Fig. 12). Of the five ligands and apoprotein studied, four ligands and apoprotein show obvious slow chemical exchange between a well-populated minor resonance and the most abundant resonance. In the case of T0070907, exchange was difficult to detect, likely because exchange rates $<\sim0.4\,\mathrm{s}^{-1}$ are too slow to be effectively detected using this method under our experimental conditions (Supplementary Fig. 12a, b).

We next quantified the exchange rate between peaks. PPARγ$^{K502C}$-BTFA bound to GW9662 and troglitazone each showed very slow exchange between peaks, making precise measurement difficult (0.3 and 0.4 s$^{-1}$; Supplementary Fig. 12d). However, for pioglitazone and ciglitazone, the exchange between the two prominent peaks is 1.2 and 1.4 s$^{-1}$ (95% confidence interval (CI): 1.1–1.3 s$^{-1}$ for pioglitazone and 1.3–1.5 s$^{-1}$ for ciglitazone; Fig. 5b). In addition, we found exchange of 1.0 s$^{-1}$ (95% CI: 0.9–1.1 s$^{-1}$) between the two resolved apo peaks of PPARγ$^{C313A,Q322C}$-BTFA (Supplementary Fig. 12c). These slow exchange rates are consistent with the notion that the detected conformations originate from larger scale movements involving many atoms[36], indicating helix 12 and the co-regulator-binding surface exchanges between two or more distinct conformations. Almost all of the signal in the ligand-bound spectra do not overlap with the apo spectrum, thus implying that exchange between conformations is occurring while bound to ligand. These data raise the possibility that these ligands are found in at least

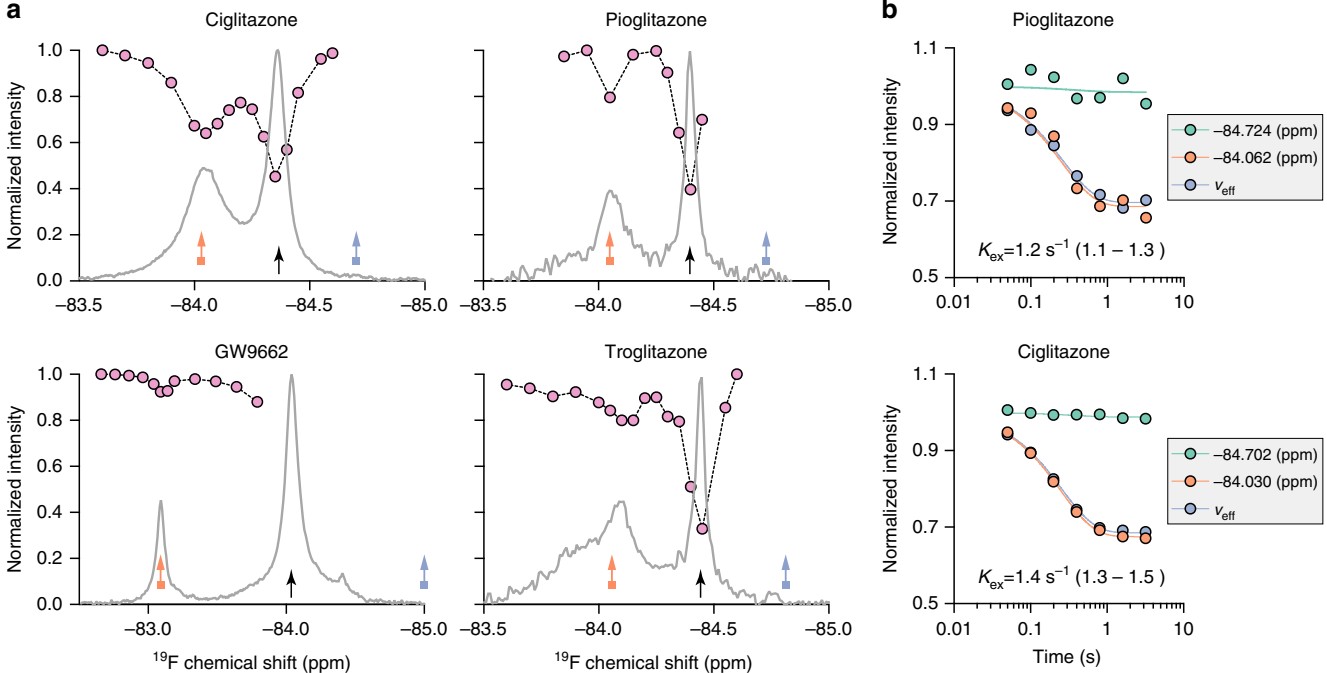

**Fig. 5** Chemical exchange saturation transfer indicates slow exchange between multiple helix 12 conformations. **a** A selective Gaussian pulse was used to saturate the $^{19}$F spectra bound to the indicated ligands at locations indicated by pink circles. The height of the pink circles indicates the height of the peak indicated by the black arrow when the selective pulse was carried out at the chemical shift of the pink circle. If exchange is occurring between the two peaks, the pink dots should mirror the spectrum. **b** A selective Gaussian pulse was used to saturate the spectrum at an on resonance (**a**; orange box arrow) and off resonance (**a**; blue box arrow) location, the peak height of the most abundant resonance (**a**; black arrow) was monitored as a function of the duration of the saturation pulse, and the resulting peak intensities were fit to extract the exchange rate. Ninety-five percent confidence intervals for the fit of the calculated exchange rates are shown within parentheses. PPARγ$^{K502C}$-BTFA was used for these data, except for ciglitazone which used PPARγ$^{C313A,}$ $^{K502C}$-BTFA for increased signal to noise. These experiments were performed once

two functionally distinct main conformations exchanging on the seconds time scale. This idea is consistent with the fact that the PPARγ non-agonist/antagonist SR1664 has been crystalized in two distinct conformations[13, 37].

**Connecting the helix 12 structural ensemble to function**. To test whether the $^{19}$F NMR detected helix 12 conformations correlate to function within our set of 16 ligands, we compared the mean-weighted $^{19}$F NMR chemical shift values to ligand efficacy for recruitment of MED1 (coactivator), NCoR (corepressor), and for a subset of these ligands, CBP (coactivator) in the TR-FRET co-regulator interaction assay. For PPARγ$^{K502C}$-BTFA there is a correlation between mean ligand-induced $^{19}$F NMR chemical shift and ligand-induced MED1, CBP, and NCoR recruitment efficacy (Fig. 6a). There are similar correlations for PPARγ$^{C313A,}$ $^{K502C}$-BTFA (Supplementary Fig. 14) but not for PPARγ$^{C313A,}$ $^{Q322C}$-BTFA. In addition, we used FP to measure the affinity of a subset of PPARγ–ligand complexes for co-regulators (MED1, NCoR, and SMRT) and compared these to labeled and wt PPARγ and found that ligand-specific affinity correlates with NMR chemical shift (Fig. 6b and Supplementary Fig. 14). The most efficacious agonist for MED1 and CBP binding in our ligand set (GW1929) induces the most upfield mean chemical shift of the PPARγ$^{K502C}$-BTFA probe. Similar trends are observed for other agonists, including the right-shifted peaks of thiazolidinedione (TZD/glitazones) ligands. In contrast, downfield mean chemical shifts are prevalent in apoprotein and inverse-agonist (T0070907)-bound PPARγ$^{K502C}$-BTFA, which show the highest efficacy and affinity for NCoR and SMRT binding (Fig. 6). These data indicate that the conformations detected by $^{19}$F NMR are functionally distinct.

**Co-regulators shift the conformational ensemble**. The data above demonstrate that ligands of different pharmacological activities can stabilize functionally distinct co-regulator-binding surface conformations. To determine how co-regulator-binding influences these ligand-dependent conformational ensembles in solution, we performed $^{19}$F NMR with and without coactivator (MED1 and CBP) and corepressor (SMRT and NCoR) peptides, and in the absence or presence of the most efficacious agonist (GW1929) or inverse agonist (T0070907) and a less efficacious inverse agonist/antagonist (GW9662) (Fig. 7a). Similar to strong agonists, addition of coactivators to apoprotein induces an upfield (right) shifting of the spectra. In contrast, similar to inverse agonists, addition of corepressors to apoprotein induces a downfield (left) shifting of the spectrum. Notably, corepressor co-binding to T0070907/PPARγ$^{K502C}$-BTFA results in smaller perturbations to the fluorine spectra than coactivator co-binding. The opposite is observed for co-regulator co-binding to GW1929/ PPARγ$^{K502C}$-BTFA. Apoprotein-bound and GW9662-bound PPARγ$^{K502C}$-BTFA are almost equally changed by the addition of coactivators or corepressors (Fig. 7c). These data suggest that the GW1929/PPARγ and T0070907/PPARγ co-regulator-binding surface structural ensembles are near ideal for MED1/CBP and NCoR//SMRT binding, respectively. In contrast, the apoprotein and GW9662 structural ensembles are not ideal for binding any of these co-regulators.

**RXRα heterodimerization shifts the inverse agonist ensemble**. PPARγ binds enhancer regions on DNA and affects gene expression primarily as a heterodimer with RXRα[38]. We therefore performed experiments to determine the effect of RXRα heterodimerization on the conformational ensemble and

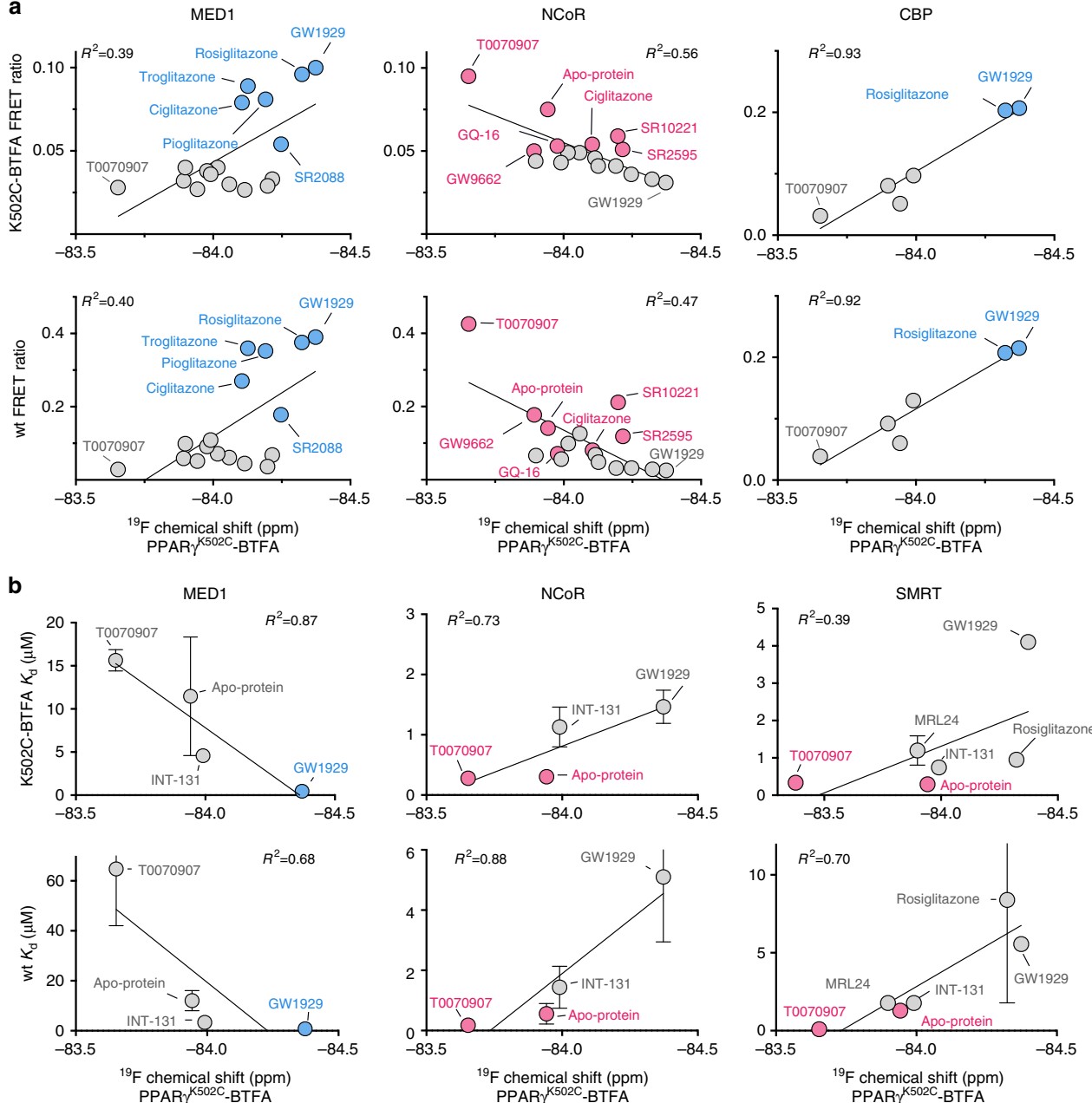

**Fig. 6** Ligand-directed helix 12 ensemble dictates PPARγ–co-regulator interaction. **a** Plot of mean $^{19}$F NMR chemical shift values (PPARγ$^{K502C}$-BTFA) vs. TR-FRET endpoint data for the recruitment of MED1, NCoR, and CBP peptides to PPARγ$^{K502C}$-BTFA (top panels) and wt PPARγ LBD (bottom panels) for the set of 16 pharmacologically distinct synthetic PPARγ ligands and apoprotein (select ligands for CBP). Error bars are relatively small for endpoint TR-FRET values (Fig. 1 and Supplementary Figure 6) and are excluded for clarity. **b** Plot of mean $^{19}$F NMR chemical shift values (PPARγ$^{K502C}$-BTFA) vs. MED1, NCoR, and SMRT peptide dissociation constant ($K_d$) for PPARγ$^{K502C}$-BTFA (top panels) and wt PPARγ LBD (bottom panels) as measured by fluorescence polarization for a subset of the ligands in **a**. Linear regression fit is shown as a solid line and the correlation coefficient ($R^2$) for the fitted line is indicated. The ligands with the highest efficacy for MED1 and NCoR recruitment for PPARγ$^{K502C}$-BTFA are highlighted. Error bars represent standard deviation of two (K502C-BTFA $K_d$ and wt SMRT $K_d$) or three (wt MED1 and NCoR $K_d$) independent experiments

on co-regulator recruitment. Other than the expected broadening and consequently decreased signal from the larger molecular weight complex, RXRα heterodimerization has a relatively small effect on the $^{19}$F spectra of PPARγ$^{C313A,K502C}$-BTFA when co-bound to GW1929 (agonist), MRL24 (less efficacious partial agonist), and apo. RXRα binding to PPARγ$^{K502C}$-BTFA co-bound to T0070907 (inverse agonist) induces a large change in the spectrum, decreasing the relative population found in the left-shifted peak (Fig. 8a), which is expected to disfavor NCoR binding. Consistent with the change in $^{19}$F NMR spectrum,

NCoR binding is decreased and MED1 recruitment is increased by RXRα heterodimerization with the T0070907/PPARγ complex (Fig. 8).

**Simulations suggest a diverse ligand-dependent ensemble**. The extreme broadening observed in apo PPARγ alone or bound to NCoR peptide originate from multiple conformations exchanging every μs to ms. We reasoned that we could sample some of these conformations in independent long time-scale simulations of apo

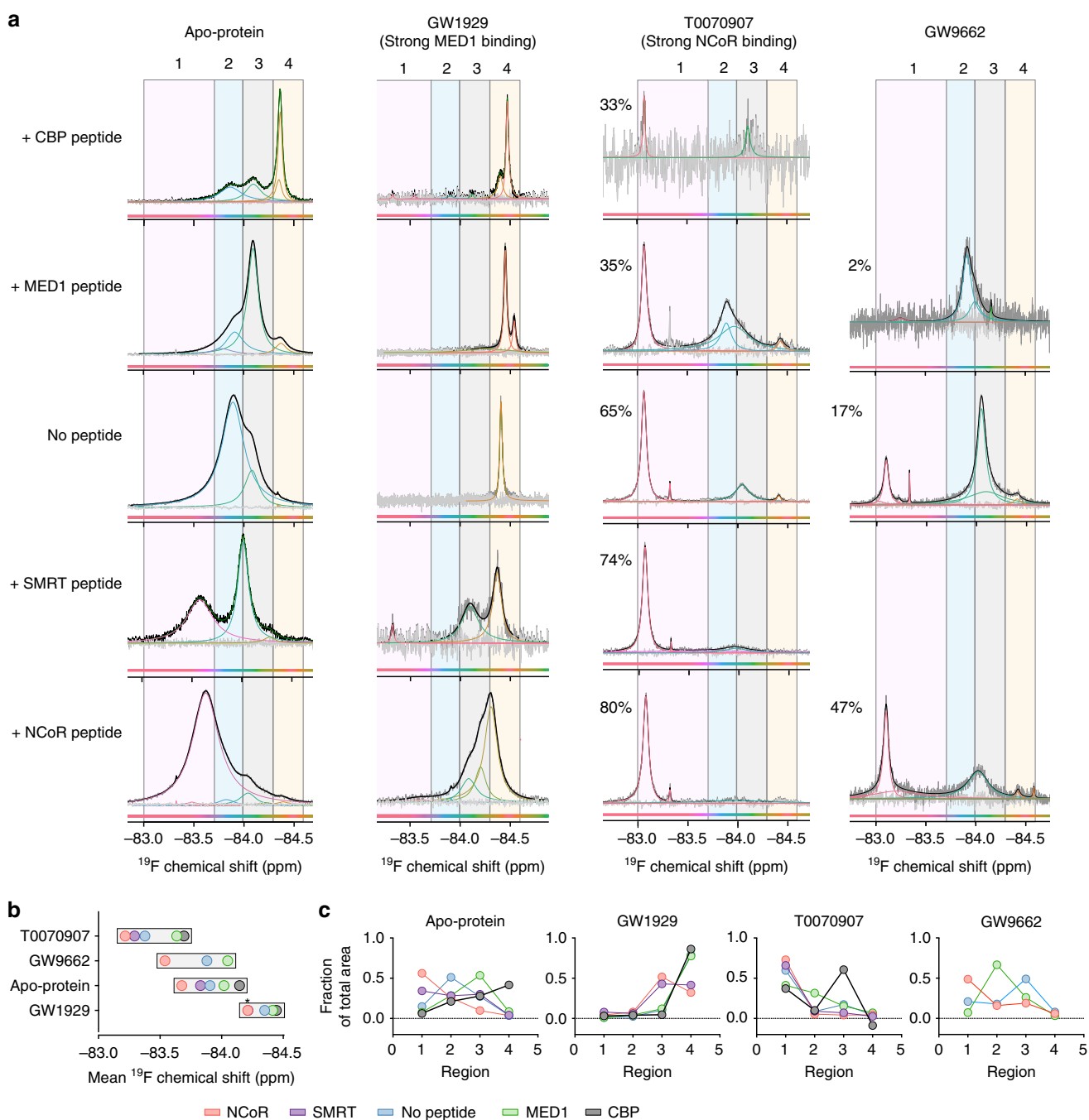

**Fig. 7** Co-regulator-binding shifts the helix 12 conformational ensemble. **a** Deconvoluted $^{19}$F NMR spectra of apo PPARγ$^{C313A,K502C}$-BTFA and PPARγ$^{K502C}$-BTFA bound to GW1929 or T0070907 in the absence and presence of MED1 coactivator or NCoR corepressor peptides. The percent of total signal area found in the left sharp peak is shown for T0070907 and GW9662 spectra. The small sharp peak at ~−83.3 ppm is free BTFA. **b** Mean-weighted chemical shift values from the plots in **a**. **c** Fraction of the total peak areas in the four colored boxed regions in **a**, which roughly correspond to the clustered spectral regions from Fig. 2; with the two middle regions corresponding to the two peaks observed for apoprotein. Agonist-bound PPARγ is changed little by MED1 binding, whereas NCoR binding changes the spectrum drastically and vice versa for inverse-agonist (T0070907)-bound PPARγ. *SMRT-induced mean chemical shift in GW1929-bound PPARγ$^{K502C}$-BTFA is the same as NCoR. These experiments were performed once

alone or apo bound to NCoR. We also simulated inverse agonist (T0070907)-bound PPARγ LBD with and without co-bound NCoR to gain additional insight into how this ligand changes the co-regulator-binding surface conformational ensemble. We built simulation models using crystal structures with helix 12 in an inactive chain B conformation (Supplementary Fig. 2d) to avoid steric clash with bound NCoR peptide. We ran simulations at 37 ° C (NMR was run at 25 °C) and with the TIP3P water model

(TIP3P is much less viscous than actual water[39]) to speed relaxation to a local or global energy minimum (i.e., stable conformation) given that this chain B-inactive conformation may be in a higher energy conformation. These simulations were allowed to run until reaching a structure that remained reasonably stable for at least 5 μs as judged by consistent helix 12 RMSD relative to the starting structure (Supplementary Fig. 15). These stable conformations are representative of a local or global free

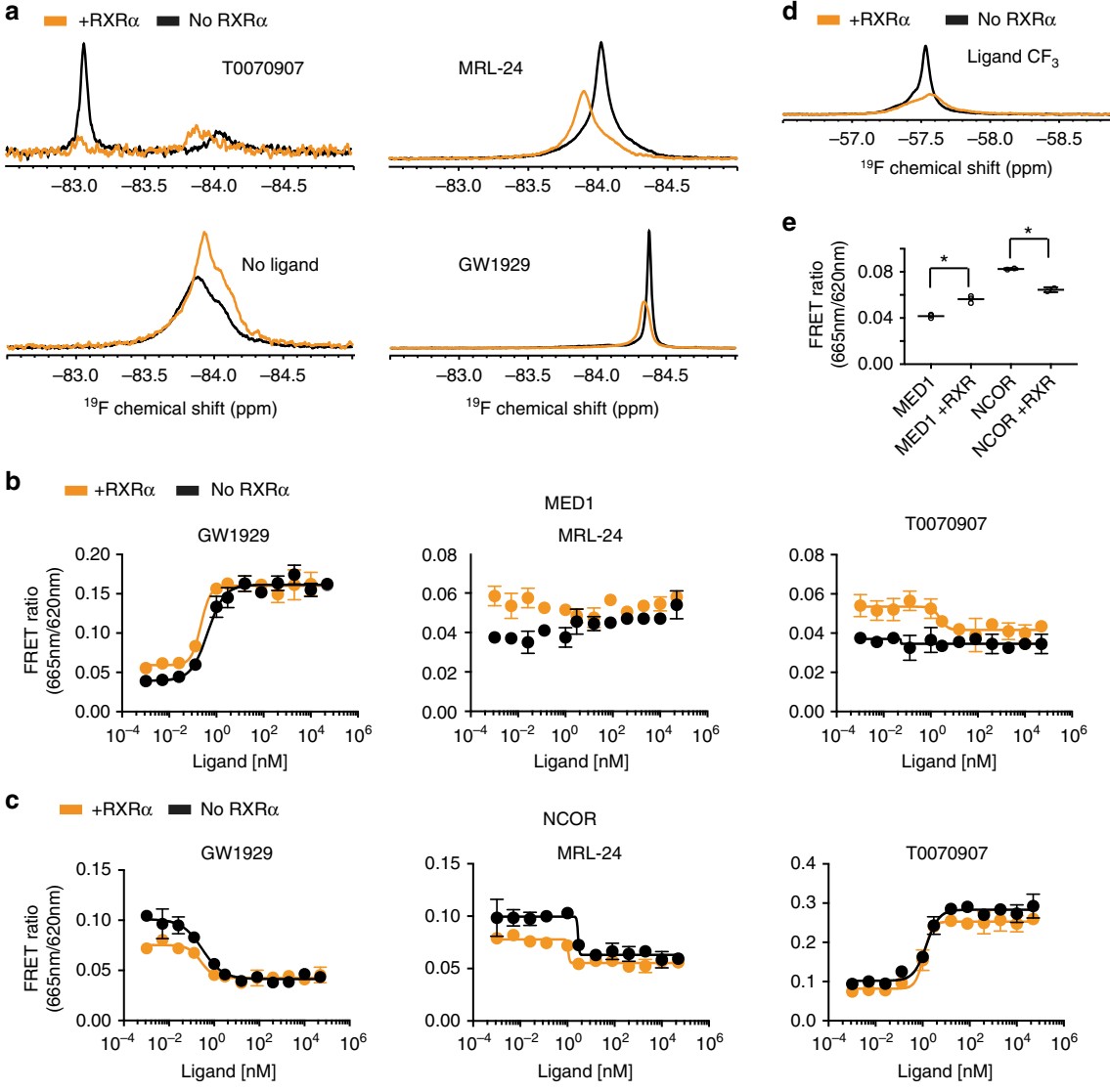

**Fig. 8** Heterodimerization of PPARγ LBD with RXRα LBD favors coactivator binding. **a** $^{19}$F NMR of PPARγ$^{K502C}$-BTFA bound to T0070907 and PPARγ$^{C313A,502C}$-BTFA bound to MRL24 or GW1929 or with no ligand bound was performed in the presence (orange) or absence (black) of RXRα LBD. These experiments were performed once. **d** The $^{19}$F NMR signal from the MRL24 ligand. Broadening and consequent reduction in signal intensity is expected as a consequence of the increased rotational correlation time of the heterodimer complex. **b**, **c** TR-FRET was used to measure interaction between wt PPARγ LBD and MED1 or NCoR in the presence or absence of equimolar concentrations of RXRα. Error bars represent standard deviation of two technical replicates within a single experiment. The experiment was repeated twice and gave similar results each time. **e** Heterodimerization favors MED1 binding ($p = 0.0017$) and disfavors NCoR binding ($p = 0.0076$) to apo PPARγ. In addition, visual and statistical comparison of NCoR and MED1 recruitment to PPARγ LBD saturated with ligand (four highest concentrations of ligands) indicates that RXRα affects co-regulator recruitment to T0070907-bound PPARγ (NCoR, $p = 0.0042$; MED1, $p = 0.0027$) more than GW1929 (NCoR, $p = 0.44$; MED1, $p = 0.34$) or MRL24 (NCoR, $p = 0.0141$; MED1, $p = 0.061$). All $p$ values are derived from a two-tailed $t$ test

energy minima. Three of four independent simulations of PPARγ LBD alone (apo) relaxed to a cluster of distinct but similar conformations that are somewhat similar to the starting chain B crystal structure, while the fourth does not appear to stabilize to the same degree and goes to a distinct conformation. These results are consistent with the broad ligand-free (apo) PPARγ $^{19}$F peaks observed which indicate exchange between two or more conformations (Fig. 9a) and are also consistent with previous free energy calculations that indicated that the inactive chain B crystal structure is very similar to low-energy apo structures[40]. Addition of NCoR to apo PPARγ LBD results in up to 12 μs of relative instability (Supplementary Fig. 15b) before finally relaxing to several different stable conformations that are consistent with the idea that NCoR pushes helix 12 away from the

co-regulator-binding surface, yielding a diverse collection of conformations that are consistent with the observed broad helix 12 $^{19}$F NMR peaks observed for this complex (Fig. 9b). All three simulations of T0070907 bound to PPARγ LBD relax to very distinct conformations (Fig. 9c). Addition of NCoR to T0070907-bound PPARγ provides a path to one dominant energy minima as all three simulations in this complex reach conformations similar to an active conformation, but with helix 12 shifted to accommodate the longer LXX I/H IXXX I/L helix of corepressors compared to the LXXLL motif of coactivators[41]. These data are consistent with our $^{19}$F NMR data that demonstrate that NCoR binding to the T0070907 PPARγ LBD complex induces a shift towards one particular conformation from several conformations (Fig. 9d).

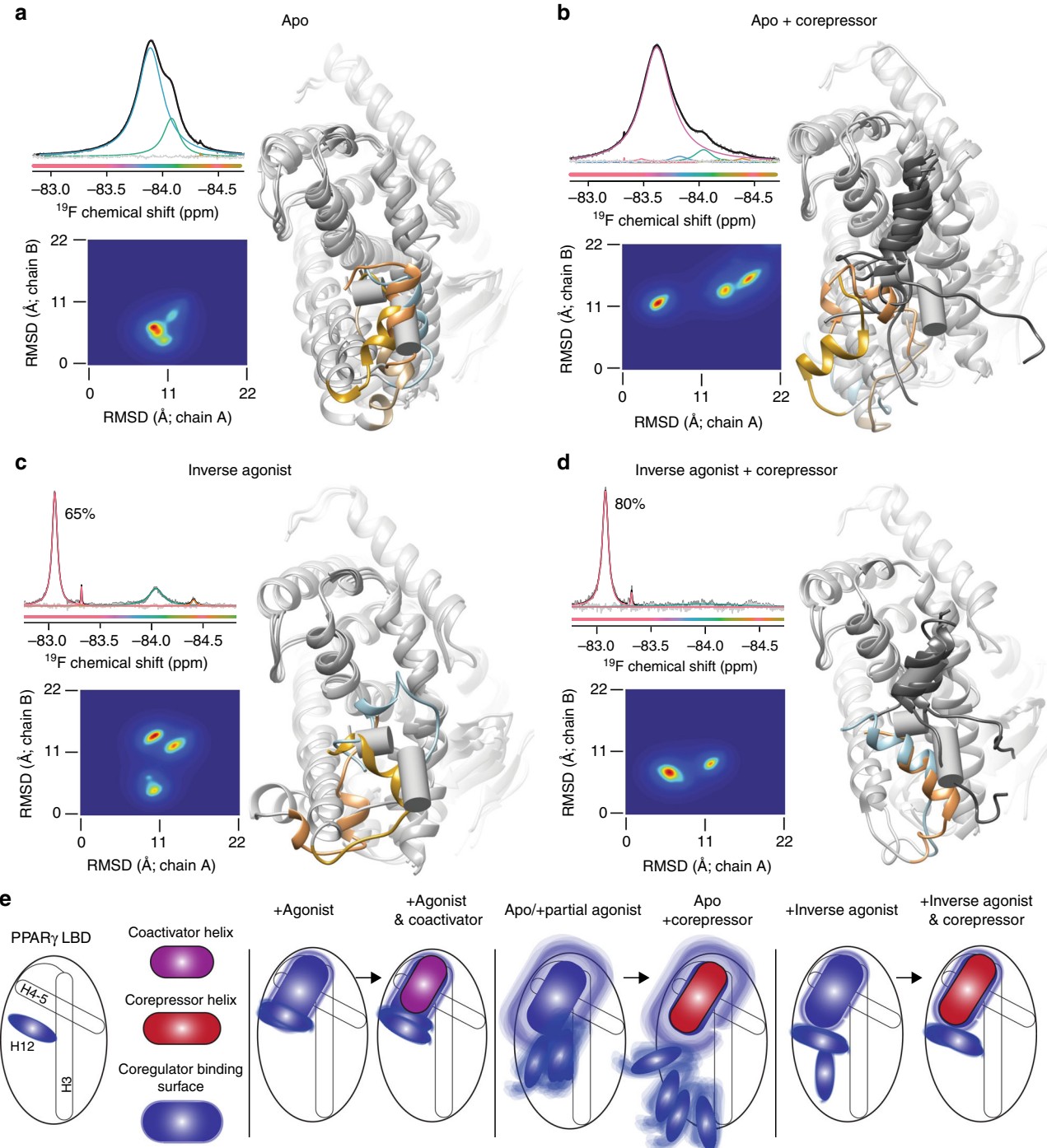

**Fig. 9** Microsecond time-scale molecular simulations point to a possible NCoR bound structure. **a–d** Root mean square deviation (RMSD) of helix 12 relative to an inactive (1PRG chain B) structure and an active conformation (1PRG chain A) for helix 12 in four different simulated complexes. Helix 12 from these inactive and active structures is shown in all panels (gray pipe). SMRT bound to PPARα (PDB code: 1KKQ) was aligned to 1PRG chain A and the SMRT helix is shown in **b** and **d** (gray pipe). Fourteen independent simulations of **a** apo PPARγ LBD, **b** apo + NCoR corepressor, **c** inverse-agonist (T0070907)-bound PPARγ LBD, and **d** PPARγ LBD co-bound to T0070907 and NCoR corepressor were carried out for 10 to 30 μs and the last 5 μs were analyzed (4 each for apo simulations and 3 each for T0070907 simulations). Colors on the RMSD plots indicate relative number of frames; red indicates the most prevalent and blue the least. Representative structures from the centroid of the major clusters are shown with helix 12 highlighted in color (4 apo, 4 apo + NCOR, 3 T0070907 and 2 T0070907 + NCoR). Helix 12 disassociates from the LBD in one apo + NCoR simulation. The cluster for this simulation (centered at 21.3 Å (x) and 23.8 Å (y)) is not shown; however, a representative structure is shown in light blue. Corresponding ¹⁹F NMR data from Fig. 7 are shown in each panel. **e** A model for helix 12 conformational diversity based on simulation and experiment. Apo PPARγ or partial-agonist-bound PPARγ helix 12 is found in many similar conformations of varying helical structure producing broad NMR peaks and rapid hydrogen deuterium exchange, while full agonist is found in a tighter cluster of conformations. Co-binding of the inverse agonist T0070907 and the corepressor NCoR produces one main conformation similar to an active conformation, but with helix 12 shifted. Fuzziness implies intermediate exchange (μs to ms) between the conformations

## Discussion

The data presented here explicitly reveal the conformational ensemble of the co-regulator-binding surface, including helix 12, of a nuclear receptor and the effects of ligands on that ensemble. A model of nuclear receptor activation arises from these data and previously published work, which indicates that nuclear receptors and other proteins are found in ensembles of structures and not a single structure[17, 22–24, 42, 43]. The structural variance around this primary structure varies from protein to protein; however, for nuclear receptors this variance appears to be considerable[16, 17, 44]. The data presented here indicate that the co-regulator-binding surface, including helix 12, exchanges relatively quickly (i.e., µs to ms lifetimes) between many conformations for apo PPARγ and partial-agonist-bound PPARγ. Agonist and inverse agonist binding consolidates this complex apo ensemble into structurally distinct active or inactive state(s) which favor coactivator or corepressor binding respectively (Figs. 7 and 9e). Very slow exchange (seconds) across high kinetic barriers is observed for some agonists and inverse agonists, indicating that these ligands hold helix 12 in narrow and deep energy wells with rare exchange occurring between sub-ensembles (Fig. 5). Consistent with these data, free energy calculations indicate that helix 12 in apo PPARγ is found in two broad energy wells with conformations similar to both active and inactive apo PPARγ helix 12 crystal conformations, whereas helix 12 in rosiglitazone-bound PPARγ is found in one deep narrow well with a conformation similar to the active chain A conformation[40]. Importantly, these data reveal a correlation between ligand efficacy and the prevalence of at least three distinct structural ensembles using a diverse set of 16 pharmacologically distinct PPARγ ligands.

The long time-scale simulations presented here are qualitatively consistent with the experimental results. Simulations started from an inactive chain B structure of PPARγ relax to multiple different structures depending on the ligand and co-regulator that are bound to PPARγ (Fig. 9). Apo PPARγ LBD relaxes to conformations similar to the starting inactive chain B structure, while NCoR binding pushes helix 12 into various different structures, consistent with the idea that NCoR does not interact productively with helix 12 in apo PPARγ. In contrast, simulations of NCoR and T0070907 co-bound to PPARγ relax to two very similar structural clusters, which are consistent the idea that helix 12 interacts productively with NCoR in this complex. In addition, simulations of GW1929 bound to PPARγ starting with helix 12 in an active chain A conformation remain in a similar helix 12 conformation throughout the simulation (Supplementary Fig. 3), indicating that as expected this is the dominant conformation for agonist-bound PPARγ in solution. These simulations also indicate that the length and helicity of helix 12 can weaken in some conformations (Fig. 9), which could contribute to broad [19]F NMR peaks and increased exchange rates in HDX-MS[13]. Overall, it is encouraging that these non-converged simulations qualitatively agree with [19]F NMR and provide a glimpse of possible conformations that comprise a portion of the observed [19]F NMR spectra; however, quantitative comparison between the [19]F NMR spectra and converged simulations remains a future challenge.

This view of the co-regulator-binding surface, including the number and relative populations of conformations and sub-ensembles that comprise the ensemble is made possible by using a single fluorine probe which provides high sensitivity, a single signal which obviates the need to transfer spin between residues, and a total lack of background signal. [19]F NMR requires mutation and labeling of PPARγ which perturbs corepressor affinity (especially SMRT) and has little effect on coactivator affinity as measured by TR-FRET peptide recruitment and FP affinities for corepressor (NCoR and SMRT) and coactivator (MED1 and CBP) peptides (Supplementary Fig. 7). Based on these observed

functional effects, we propose that any effect from the mutations would likely shift inverse-agonist-bound mean NMR chemical shifts toward the center decreasing the population found in the inverse agonist conformational state (cluster 3). Depending on the chemical shift difference and rate of exchange between the conformations this will result in movement of a peaks chemical shift and/or a decrease in population of the cluster 3 (left cluster). For example, a left-shifted narrow peak at a chemical shift typical of inverse agonism (cluster 3) and a broad peak at a chemical shift typical of antagonist/partial agonist (cluster 2) are observed for PPARγ[K502C]-BTFA bound to the inverse agonist T0070907[29, 30] and less efficacious inverse agonist/antagonist GW9662 (Figs. 2 and 7). Addition of NCoR or SMRT shifts the population from cluster 2 to cluster 3, while the mutations and labeling may shift the equilibrium population in the opposite direction toward the antagonist/partial agonist cluster (cluster 2). Thus, it may be that helix 12 of wt PPARγ LBD bound to T0070907 is found in a conformation represented by the inverse agonist cluster (cluster 3; left-shifted narrower peak) to a larger degree than detected by PPARγ[K502C]-BTFA.

This work adds detail to how ligands in general control the activity of nuclear receptors. Our data not only confirm that helix 12 and the co-regulator-binding surface exist as a ligand-specific dynamic structural ensemble but also reveal the relative populations of sub-ensembles that comprise the overall structural ensemble and correlate function with this ensemble. Further definition of the conformational ensemble of the entire protein and the kinetics and thermodynamics of exchange between the members of the ensemble will build an accurate model of how ligands produce functional outputs via nuclear receptors and allow greater control of their function via ligands.

## Methods

**Protein purification.** A pET-46 plasmid carrying the genes for ampicillin resistance and N terminally 6xHis-tagged PPARγ containing a tobacco etch virus (TEV) nuclear inclusion protease recognition site between the His tag and protein of interest was transformed into chemically competent E. coli BL21(DE3) Gold cells (Invitrogen). Cells were grown in TB at 37 °C were induced at an OD$_{600}$ of approximately 0.8 by the addition of 0.5 mM isopropyl β-D-1-thiogalactopyranoside (IPTG) and the temperature lowered to 22 °C. Induction proceeded for 16 h prior to harvesting. Harvested cells were homogenized into 50 mM phosphate (pH 8.0), 300 mM KCl, 1 mM tris(2-carboxyethyl)phosphine (TCEP), and lysed using a C-5 Emulsiflex high-pressure homogenizer (Avestin). Lysates were then clarified and passed through two Histrap FF 5 ml columns in series (GE Healthcare). Protein was eluted using a gradient from 15 to 500 µM imidazole. Fast protein liquid chromatography was performed on either an NGC Scout system (Bio-Rad) or an ÄKTA Start (GE Healthcare). Eight milligrams of recombinant 6xHis-tagged TEV was added to eluted protein followed by dialysis into 50 mM Tris (pH 8.0), 200 mM NaCl, 1 mM TCEP, and 4 mM EDTA. The protein was again passed through HisTrap FF columns in order to separate cleaved protein from TEV as well as the cleaved 6xHis tag. The cleavage step was only performed on protein which would be used for NMR or FP, but the protein used for TR-FRET did not have the 6xHis tag removed. The protein was then further purified by gel filtration using a HiLoad 16/600 Superdex 200 PG (GE Healthcare). Size exclusion was performed in 25 mM MOPS (pH 8.0), 300 mM KCl, 1 mM TCEP, and 1 mM EDTA buffer. Protein was then dialyzed into 25 mM 3-(N-morpholino)propanesulfonic acid (MOPS) (pH 7.4), 25 mM KCl, and 1 mM EDTA buffer. Protein purity in excess of 95% was determined by gradient 4–20% sodium dodecyl sulfate-polyacrylamide gel electrophoresis analysis (NuSep). Protein concentration was determined using ε$_{280}$ = 12,045 M$^{-1}$ cm$^{-1}$.

[15]N-labeled protein was grown in M9 minimal media containing 99% [15]NH$_4$Cl (Cambridge Isotope Laboratories) as the sole nitrogen source. For this growth, cells were grown at 37 °C and 180 rpm until an OD$_{600}$ of approximately 1.0 was reached. At this point, the temperature was dropped to 22 °C for 1 h. Following cool down period, protein expression was induced by the addition of 500 µM IPTG during which induction cells remained at 22 °C. Protein expression and purification was then accomplished utilizing the same protocol as outlined above.

**Delipidation of PPARγ.** To delipidate PPARγ LBD, purified protein was diluted to 0.8 mg ml$^{-1}$ and batched with Lipidex 1000 (Perkin-Elmer) at an equal volume. This mixture was batched for 1 h at 37 °C and 100 rpm. Immediately following this treatment, protein was pulled through a gravity column by syringe. To increase

yield, it was found that the speed of elution was important; protein could not remain on the resin at room temperature in excess of 3 min. Two more column volumes of pre-warmed 25 mM MOPS, 25 mM KCl, and 1 mM EDTA were also pulled through in the same manner. Quality of delipidation was then estimated by $^{19}$F NMR, and loss of lipid can be most easily detected by a reduction in the peak at −84.1 ppm. When non-delipidated protein is used in this report it is labeled as bound to *E. coli* lipids.

**Site-directed mutagenesis**. Mutations in PPARγ LBD were generated using the Quikchange Lightning site-directed mutagenesis kit (Agilent) using primers listed in Supplementary Table 6. The presence of expected mutations and absence of spurious mutations was confirmed by Sanger sequencing (Eurofins).

**Preparation of NMR samples**. NMR samples were prepared to a final concentration of 150 μM protein in 470 μL volume containing 10% D$_2$O. Addition of ligand was done in two separate injections of compound to reduce precipitation. Injections were spaced 30–60 min apart to allow time for binding. All ligands were dissolved in D$_6$-dimethylsulfoxide (DMSO), with the exception of GQ-16, which was dissolved in D$_7$-dimethylformamide (DMF). Deuterated solvents were obtained from Cambridge Isotope Laboratories Inc. and were at least 99% iso-topically pure. Final concentrations of ligand for samples of PPARγ$^{K502C}$-BTFA were 1.25x ligand to protein (187.5 μM), with the exception of troglitazone, pio-glitazone, and ciglitazone, which were loaded to 2.0x (300 μM) due to poor binding affinity. In samples of PPARγ$^{C313A,K502C}$-BTFA, ligand concentration was 1.1x to protein (165 μM), with the exception of troglitazone, pioglitazone, and ciglitazone, which were loaded to 1.5x (225 μM). To decrease the likelihood of labeling the single native cysteine (C313) in PPARγ$^{K502C}$-BTFA, we first loaded ligands into the protein and then labeled with BTFA since bound ligands would restrict access to C313 for all experiments involving PPARγ$^{K502C}$-BTFA. In both cases, the ligand concentration in DMSO was controlled to maintain a constant volume of DMSO or DMF addition to the sample (8.80 μL for PPARγ$^{K502C}$-BTFA and 7.76 μL for PPARγ$^{C313A,K502C}$-BTFA) including for apo and *E. coli* lipid samples. For peptide studies, 1 mM peptide in identical buffer to protein was added at a 2:1 molar ratio (final concentration 300 μM peptide and 150 μM protein). For non-covalent ligands, samples were labeled with 2.0x BTFA after the addition of ligand. For covalent ligands, samples were labeled with 10x BTFA following preincubation of T0070907 or GW9662; and addition of 1.5 and 2 ligand molar ratios yielded very similar spectra, indicating complete covalent modification of C313 and likely no bonding to K502C (Supplementary Fig. 5). After the addition of BTFA, protein was incubated for 30–60 min and then buffer exchanged at least 100x using 10 kDa Amicon Ultra-15 concentrators (Merck Millipore) to remove excess unbound BTFA. Following this buffered D$_2$O was added.

Variable D$_2$O NMR samples were prepared by buffer exchanging protein samples >100x into 25 mM MOPS, 25 mM KCl, 1 mM EDTA (pD 7.4) buffer prepared in either 50 or 100% D$_2$O using Amicon Ultra-4 10 kDa centrifugal filters (EMD Millipore). One hundred percent D$_2$O buffer was adjusted to read pD 7.4 (pH 7.0[45]), and then mixed with appropriate amounts of H$_2$O buffer that had been adjusted to read pH 7.4. Following this, ligands were added to the appropriate concentration as usual. The samples which contained only 10% D$_2$O were prepared as usual with the standard addition of 10% buffered D$_2$O to the final sample. Samples of PPARγ$^{K502C}$-BTFA were already loaded with ligand and labeled appropriately with BTFA prior to exchange into deuterated buffers.

**FP assay**. FP peptide binding assays were performed by plating a mixture of 50 nM peptide with an N-terminal FITC tag, 12-point serial dilutions of PPARγ-LBD (wt, PPARγ$^{K502C}$-BTFA, PPARγK502C, or PPARγ$^{C313A,K502C}$-BTFA), and PPARγ ligands from 50 μM to 24 nM. PPARγ-LBD and PPARγ ligands were added at a 1:1 ratio. This mixture was added to wells of low-volume 384-well black plates (Grenier Bio-one, catalog number 784076) to a final volume of 16 μL. Peptides were synthesized by Lifetein LLC (Somerset, NJ, USA) for the for MED1 peptide, sequence: NTKNHPMLMNLLKDNPAQD; and the NCoR peptide, sequence: GHSFADPASNLGLEDIIRKALMG (2251–2273). Other peptides were purchased from ThermoFisher (Waltham, MA, USA) for MED1 peptide, sequence: NTKNHPMLMNLLKDNPAQD (catalog number PV4549); CBP peptide, sequence: AASKHKQLSELLRGGSGSS (catalog number PV4596); and SMRT, sequence: HASTNMGLEAIIRKALMGKYDQW (catalog number PV4424). All dilutions were made in 25 mM MOPS (pH 7.4), 25 mM KCl, 1 mM EDTA, 0.01% fatty-acid-free bovine serum albumin (BSA) (EMD Millipore, catalog number 126575), 0.01% Tween, and 5 mM TCEP. Assay titrations were performed in duplicate. Plates were incubated in the dark at room temperature for 2 h before being read on a Synergy H1 microplate reader (BioTek). FP was measured by excitation at 485 nm/20 nm and emission at 528 nm/20 nm for FITC. Data were fit using nonlinear regression (agonist vs. response – variable slope 4 parameters) in Prism 7.0b. For FP, TR-FRET, and Fluormone competitive binding assays, we did two technical replicates and repeated these experiments independently in the lab for once (Fluormone) or 2 or more times FP and TR-FRET. We chose these number of technical replicates and independent experiment replicates based on our experience with the limited variability inherent in these biochemical assays

**Time-resolved Förster resonance energy transfer assay**. TR-FRET peptide recruitment assays were performed by plating a mixture of 8 nM 6xHis-PPARγ-LBD (wt, PPARγ$^{K502C}$-BTFA, or PPARγ$^{C313A,K502C}$-BTFA), 0.9 nM LanthaScreen Elite Tb-anti-His antibody (LifeTechnologies catalog number PV5863), 200 nM peptide (N terminally biotinylated and C terminally amidated), 400 nM streptavidin-d2 (Cisbio, catalog number 610SADLB), and 12-point serial dilutions of PPARγ ligands from 50 μM to 1 pM. This mixture was added to wells of low-volume 384-well black plates (Grenier Bio-one, catalog number 784076) to a final volume of 20 μL. Peptides were synthesized by Lifetein LLC (Somerset, NJ, USA) for MED1 peptide; sequence: VSSMAGNTKNHPMLMNLLKDNPAQ; and NCoR peptide, sequence: GHSFADPASNLGLEDIIRKALMG (2251–2273). All dilutions were made in 25 mM MOPS (pH 7.4), 25 mM KCl, 1 mM EDTA, 0.01% fatty-acid free BSA (EMD Millipore, catalog number 126575), 0.01% Tween, and 5 mM TCEP. Assay titrations were performed in duplicate. Plates were incubated in the dark at room temperature for 2 h before being read on a Synergy H1 microplate reader (BioTek). TR-FRET was measured by excitation at 330 nm/80 nm and emission at 620 nm/10 nm for terbium and 665 nm/8 nm for d2. Change in TR-FRET was calculated by 665 nm/620 nm ratio.

Data were fit using nonlinear regression (agonist vs. response – variable slope 4 parameters) in Prism 7.0b. Outliers were automatically detected using Prism 7.0b's implementation of the ROUT method[46] and excluded from the curve fitting (20 out of 2260 total data points were flagged as outliers). However, all data points, including detected outliers, were included in figure graphs. In cases where curve fitting failed, the TR-FRET value nearest the calculated free ligand concentration in the NMR experiment was used; otherwise, the TR-FRET value at the calculated free NMR ligand concentration was calculated using the fitted curve. These TR-FRET ratio values were used to correlate with NMR chemical shift values. One ligand, nTZDpa, showed a biphasic TR-FRET curve. This could be due to several factors including ligand aggregation[47], absorption interference in the TR-FRET assay (nTZDpa contains an indole group that could in principle cause interference), or alternate site binding effects[18]; we therefore utilized the value of the TR-FRET ratio for nTZDpa based on a fit that did not include the last two concentrations (10 and 50 μM), so as to exclude most effects of the second transition. In addition, the 50 μM SR2088 point was not run because we did not have sufficient ligand. All error bars are standard deviation.

**Fluormone competitive binding assays**. PPARγ ligand inhibition constants ($K_i$) were measured using a protocol adapted from LanthaScreen TR-FRET PPARγ competitive binding assay (Invitrogen, catalog number PV4894). Assay was performed by plating a mixture of 8 nM 6xHis-PPARγ-LBD, 2.5 nM LanthaScreen Elite Tb-anti-His antibody, 5 nM LanthaScreen Fluormone Pan-PPAR Green (Invitrogen, catalog number PV4896), and 12-point serial dilutions of PPARγ ligands from 50 μM to 140 fM. This mixture was added to wells of low-volume 384-well black plates (Grenier Bio-one) to a final volume of 16 μL. All dilutions were made in 25 mM MOPS (pH 7.4), 25 mM KCl, 1 mM EDTA, 0.01% fatty-acid free BSA (EMD Millipore), 0.01% Tween, and 5 mM TCEP. Assay titrations were performed in duplicate. Plates were incubated in the dark for 2 h at room temperature before being read on a Synergy H1 microplate reader (BioTek). TR-FRET was measured by excitation at 330 nm/80 nm and emission at 495 nm/10 nm for terbium and 520 nm/25 nm for Fluormone. Change in TR-FRET was calculated by 520 nm/495 nm ratio. Nonlinear curve fitting was performed using Prism 7.0b (Graphpad Software Inc.) as described above for the TR-FRET data, including manual exclusion of highest two concentrations for nTZDpa. Thirty of the 1224 total data points for all three proteins (wt, PPARγ$^{K502C}$-BTFA, and PPARγ$^{C313A,K502C}$-BTFA) were automatically excluded by Prism in the fits.

**$K_i$ calculation**. The inhibition constant for each PPARγ ligand was calculated by applying a corrected Cheng-Prusoff[48, 49]

$$K_i = \frac{(Lb)(IC_{50})(K_d)}{(Lo)(Ro) + Lb(Ro - Lo + Lb - K_d)}, \tag{1}$$

where IC$_{50}$ is the concentration of the ligand that produces 50% displacement of the Fluormone tracer, Lo is the concentration of Fluormone in the assay (5 nM), and $K_d$ is the binding constant of Fluormone to wt or the two BTFA-labeled mutants, Ro is the total receptor concentration, and Lb is the concentration of bound Fluormone in the assay with no addition of test ligand. The affinity of Fluormone for the two BTFA-labeled mutant proteins was determined via TR-FRET by titration of Fluormone into each mutant bound to Elite Tb-anti-His antibody. Dissociation constants of Fluormone for wt was measured as 7.9 ± 0.2 for PPARγ LBD, and the variants were measured as 26 ± 3 nM for PPARγ$^{K502C}$-BTFA and 44 ± 4 nM for PPARγ$^{C313A,K502C}$-BTFA and 12 ± 1 nM PPARγC313A.

**NMR spectroscopy**. Acquisition of spectra was performed using a Bruker 700 MHz NMR system equipped with a QCI-F cryoprobe. Chemical shifts were calibrated using an internal separated KF reference in 20 mM KPO$_4$ (pH 7.4) and 50 mM KCl contained in a coaxial tube inserted into the NMR sample tube. KF was set to be −119.522 ppm, which is the shift of the KF signal with respect to the $^{19}$F basic transmitter frequency for the instrument (658.8462650 MHz) at 298.2 K, the temperature at which samples were run. Routine 1D fluorine spectra were acquired

utilizing the zgfhigqn.2 pulse program (Bruker Topspin 3.5), which consists of a 90° pulse followed by acquisition with proton decoupling (acquisition = 0.7 s). Settings were D1 = 1.2 s, AQ = 0.82 s. Approximately 500 to 4000 transients were collected. Saturation transfer experiments were carried out using the stddiff pulse program. Settings were D1 = 1.6 s, AQ = 0.6 s. For some experiments, the total duration of the saturating pulse (Gaus1.1000, 54.52 dB, 50 ms) was 1.6 s ($D_2O$) and the location of the saturating pulse was varied. In other experiments, the duration of the saturating pulse was varied, with the saturating pulse location held constant and the rate of exchange was fit using equation 50 found in a previous publication[50]. An off resonance selective saturating pulse was used to determine the peak intensity at time $t = 0$ and $R_1$ was determined experimentally (Supplementary Table 5). Fits of the saturation transfer data were accomplished with a single free parameter using these experimentally determined values for initial intensity and $R_1$ (2.7 s$^{-1}$ used for all fits except GW9662 which used 2.4 s$^{-1}$). Transverse and longitudinal relaxation lifetimes ($T_1$ and $T_2$) were determined by fitting data acquired using an inversion recovery experiment (Bruker pulse program t1ir) and cpmg pulse sequence (Bruker cpmg) in Prism 7.0b (Graphpad Software Inc.) using standard formulas. Spectra were deconvoluted in an objective manner with models chosen statistically by a fitting program[32]. All fits were carried out in the same manner with the same settings in the fitting program, except where noted. Relative phase of fitted peaks was allowed to vary slightly ($\pi/50$ rad) to accommodate imperfect phasing of these broad signals. The fitting algorithm[32] assumes Lorentzian lineshapes of similar phase. Intermediate exchange effects and field inhomogeneity are likely present in some of these spectra, which will result in inaccuracies in the fitted models; however, notwithstanding these limitations, the deconvolution method provides an objective view of the possible underlying spectral structure and populations. Two-dimensional [$^1$H,$^{15}$N]TROSY-HSQC NMR data were obtained using the trosyf3gpphsi19.2 pulse program. Select NMR spectra were replicated in two different ways: (1) Some NMR samples were measured via NMR initially and then days to weeks later to determine if certain parts of the spectrum changed. Any changes would indicate that non-reversible processes contribute to that part of the signal, such as unfolding or degradation of the protein. (2) Some spectra were run twice utilizing protein from the same batch as utilized for the first spectra or from an entirely different protein preparation.

**Molecular dynamics simulations**. Residues in our physical protein constructs used in NMR that were missing in crystal structures (except the N-terminal glycine, which is an unnatural vestige of the cleaved His tag) were added using the modeler[51,52] within Chimera[53] and a PDB file was saved. This PDB file was then submitted to the h++ server[54] (http://biophysics.cs.vt.edu/H++) to determine the state of titratable protons at pH 7.4, along with more realistic rotamers for some residues. This h++ PDB file was then given AMBER names for the various protonation states of histidine determined by h++ using pdb4amber (AmberTools14[55]). PDB files of cysteine-BTFA residue was created through modification of a cysteine residue in chimera. This PDB file was then submitted to the RED server[56] for RESP[57] charge derivation and geometry optimization. RESP values for the cysteine backbone of these modified residues were constrained to match AMBER cysteine residue values as part of the input to the RED server. In a manner similar to that outlined in sections 2–3 in tutorial 5 on the ambermd.org site (http://ambermd.org/tutorials/basic/tutorial5/), the output mol2 file was then used to prepare an ac file and then a prepin file containing the same RESP-derived charges (Supplementary Note 2) and two force modification files. The AMBER parameter database derived frcmod file (Supplementary Note 3) was loaded after the GAFF[58] parameter database derived frcmod file (Supplementary Note 4) within Tleap in order to use AMBER parameters where possible for the cysteine-BTFA residue. Tleap was then used to generate parameter and coordinate files using both ff14SB[59] and GAFF values. A truncated octahedron solvation cell with boundaries at least 10 Å from any protein atom was built with TIP3P[60] water. The system was neutralized with Na$^+$ ions and K$^+$ and Cl$^-$ atoms were added to 50 mM. Joung and Cheatham[61] ion parameters were used. Minimization (imin = 1) and equilibration was carried out in nine steps with non-bonded cutoff (cut) set to 8 Å and with the equilibrations carried out at 310 K. First, steepest descent minimization (ntmin = 2) with strong restraints (restraint_wt = 5 kcal mol$^{-1}$ Å$^{-2}$) on protein heavy atoms for 2000 steps was used followed by NTV MD with shake, the same restraints and 1 fs steps for 15 ps. Next, two rounds of 2000 steps of steepest descent minimization with progressively relaxed restraints (restraint_wton protein heavy atoms for 2000 steps was used followed by 2 and 0.1 kcal mol$^{-1}$ Å$^{-2}$) followed by a round without restraints. This was followed by three rounds of NTP MD with shake (5, 10, and 10 ps in duration), and protein heavy atom restraints of 1, 0.5, and 0.5 kcal mol$^{-1}$ Å$^{-2}$. A final unrestrained NTP MD simulation was then run for 200 ps with 2 fs steps. The final restart file from this process was used along with a hydrogen mass repartitioned parameter file (modified using parmed) to run new simulations with new randomized atomic velocities using 4 fs steps at 310 K. Analysis was carried out using CPPTRAJ[62]. All production simulations were carried out using pmemd.cuda or pmemd.cuda.MPI.

The S enantiomer of GW1929, which was used in NMR and TR-FRET, was built using Maestro (Schrödinger LLC). The pyridine ring nitrogen of GW1929 was the only atom with a predicted p$K$a near 7.4 (calculated 7.56 ± 1.12) using the EPIK module (Schrödinger LLC). We chose to model this nitrogen as deprotonated. There is no crystal structure for GW1929 bound to PPARγ; however, there is

crystal structure for GI262570, which is bound to PPARγ LBD in the 1FM9 crystal structure. GI262570 is identical to GW1929 for about 2/3 of the molecule. GW1929 was docked into the 1FM9 crystal structure with GI262570 removed using AutoDock Vina[63]. The best scoring docked binding mode overlaid well with GI262570. In this docked model in the helix 12-interacting region, the two ligands themselves are identical. RESP charges for GW1929 were derived using the RED server and force modification files were generated using GAFF parameters. 1FM9 was modified (or not) to incorporate cysteine-BTFA (parameterized as described above) in place of K502 and docked with GW1929 in a similar way to that described above to build PPARγK502C-BTFA and PPARγ bound to GW1929.

1PRG chain B was used to create the build for apo and 3BOR chain B was used to build T0070907-bound PPARγ LBD. 3BOR contains GW9662 which differs from T0070907 by one atom. T0070907 has a nitrogen in place of a carbon atom in one of the ligand rings. This change was made in chimera and the ligand parameterized and incorporated into the structure as described above for BTFA. NCoR (same sequence as used in NMR and TR-FRET including N-terminal acetylation and C-terminal amidation) was added to these builds utilizing chimera using the following procedure. The core helix structure from NCoR (from 2OVM) was aligned to SMRT on a PPARα SMRT structure (1KKQ), and then apo PPARγ chain B (1PRG) was aligned to PPARα and the PPARα/SMRT structure deleted leaving the aligned NCoR on apo PPARγ chain B. A similar procedure was used with the 3BOR structure to create T0070907 co-bound with NCoR on PPARγ. These PDBs were then used to create the final solvated, minimized, and equilibrated structure in a manner similar to that described above.

All simulations were run with settings shown in Supplementary Note 1 including utilization of SHAKE[64], with variability in the frequency of writing various files to disk. K-means clustering was performed and representative structures from these structures were outputted by CPPTRAJ[62].

**Data availability**. Data files for TR-FRET, Fluormone competitive binding assays, and FP are publically available at https://osf.io/rqdpz/. Any other datasets generated during and/or analyzed during the current study are available from the corresponding author on reasonable request.

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

## Acknowledgements

We thank James Aramini at the City University of New York Advanced Science Research Center (CUNY ASRC) for assistance in setting up the saturation transfer difference, cpmg, and t1ir experiments and Daniel R. Roe (NIH Laboratory of Computational Biology, NHLBI) for providing the minimization and equilibration script and simulation advice. NMR data presented herein were collected at the CUNY ASRC Biomolecular NMR Facility. This work was supported in part by National Institutes of Health (NIH) grants K99DK103116 (to T.S.H.), R00DK103116 (to T.S.H.), P20GM103546 (to T.H.S.), DK101871 (to D.J.K.), and DK105825 (to P.R.G.); and National Science Foundation (NSF) award 1359369 (PI Karbstein) that funds the SURF program at Scripps Florida.

## Author contributions

T.S.H. and D.J.K. conceived the study and designed experiments. I.M.C. and M.D.N. prepared samples and performed experiments. T.S.H., I.M.S.dV., J.S., and J.F. prepared samples and performed preliminary experiments. Z.H. and T.S.H. built, performed, and analyzed the molecular simulations. Y.L. assisted with preliminary experiments. A.-L.B., Y.S., and T.M.K. synthesized ligands. H.R.-C. cloned and verified K502C plasmid. P.R.G. contributed to conception, provided key reagents, and characterization of ligand pharmacology. R.G.-M. and T.E.C. III assisted T.S.H. with molecular dynamics simulations and ligand parameterization. T.S.H. and D.J.K. analyzed data and wrote the manuscript with input from all authors.
