## [Peer Review File · Nature Communications]

Reviewers' comments:

Reviewer #1 (Remarks to the Author):

The activity of nuclear receptors is regulated by ligands. Binding of ligands to the LBD of nuclear receptors affects the interactions between the AF-2 domain of the LBD (in helix 12) and transcriptional coregulators (i.e., coactivators and corepressors). There are various ligands (i.e., full and partial agonists, antagonists, and inverse agonists), coactivators and corepressors. Different ligands generate different transcriptional outcomes, although crystal structures revealed that the conformations of helix 12 are similar in LBD bound to different ligands. In an attempt to understand the structural mechanism responsible for the diverse activity of different ligands, the authors hypothesize that the helix 12 of LBD consists of a dynamic ensemble of conformations

(that provide binding surface for coregulators) in response to different ligands. To test the hypothesis, the authors use PPAR γ LBD as a model, and 19F NMR, TR-FRET assay and molecular simulations as methodology. They utilized a short peptide containing one receptor-interacting motif from one coactivator (MED1) or one corepressor (NCoR), and a collection of 16 PPAR γ ligands to facilitate their studies. Based on the results from their studies, the authors claim that different ligands induce different coregulator binding surfaces: multiple thermodynamically accessible conformations (for partial agonists or no ligand) and one thermodynamically accessible conformations (for full agonists and inverse agonists). These data provide novel information helpful for understanding the different effects of different ligands, which are of interest to readers in the nuclear receptor research community, and the wider transcription field.

Addressing the following points will strengthen the author's conclusions:

1. The authors have utilized a short peptide from MED1 or NCoR for both the TR-FRET and NMR studies. At least one additional coactivator (e.g., PGC-1) and one additional corepressor (e.g., SMRT) should be used for representative ligand from each ligand type;
2. In the TR-FRET assay (Figure 1), the basal activity (i.e. activity from solvent control; what is the solvent used to solubilize compound? DMSO?) should be included and clearly indicated (although the activity at the lower compound concentrations might be close to the basal activity);
3. The authors noticed that the labeled proteins (PPAR γ K502C-BTFA and PPAR γ C313A,K502C-BTFA) have lower binding affinity to ligand and reduced ligand-induced recruitment of MED 1 and NCoR. While Pioglitazone is a full agonist for the wild-type PPAR γ (Figure 1, for MED1), it appears to function as a partial agonist for PPAR γ K502C-BTFA (Supplementary Figure 6a, for MED1). Please discuss how the slightly different ligand behavior between the wild-type/unlabeled and mutated/labeled PPAR γ might have affected the conclusion drawn from the NMR studies.
4. Figure 6 correlates data from the NMR to that from TR-FRET for the labeled PPAR γ -BTFA. Related to point #3 above, it might be useful to include the TR-FRET data for the wild-type PPAR γ (i.e., data from Figure 1) in the comparison.
5. PPAR γ functions as a heterodimer with RXR. Please discuss whether and how the conclusions from this study might be affected if RXR is included in the assays?
6. Page 25: the last sentence of the first paragraph appears to be incomplete.
7. Figure 1: does the color arrow indicate ligand potency? If so, the position of Troglitazone and Pioglitazone should be switched; similarly, for Supplementary Figure 6a, the list of ligands on the right can be better arranged to match the potency indicated by the curves (e.g., for MED1, Pioglitazone is listed on the top, but it is not the most potent ligand revealed by the curves).

Reviewer #2 (Remarks to the Author):

The manuscript by Chrisman et al. is a novel application of fluorine NMR spectroscopy for deciphering the conformational landscape of a nuclear receptor. The authors use 16 pharmacological inhibitors as well as native coactivator and repressor ligands to provide a rich data

set to assess the different modes of activation. Due to the highly responsive fluorine nucleus, ^{19}F NMR is becoming increasingly popular for addressing difficult questions in structural biology. The current application to a nuclear receptor is of significant biomedical importance, and the authors exploited from a simple resonance, both the chemical shift and D_2O dependence, and chemical exchange experiments by fluorine NMR combined with molecular dynamics and FRET experiments to build a compelling picture of PPAR γ conformational dynamics. The conclusions in general seem sound. Although the experiments were well done, there were sections where clarity could be improved. For example, there are a variety of qualitative statements made when quantitative would be more appropriate. Uses of the phrases "well-functioning", "similar patterns", "lower potency", "some reduction of affinity" "around 1/s" could be strengthened if actual values were discussed rather than burying the values in a supporting information table. In some cases this seems to be smoothing over potential perturbing effects from fluorine labeling. For example in Figure S6 the correlation plots don't seem to be one-to-one, so as stated it is hard to interpret what similar patterns would be that give rise to similar function. Other minor concerns are listed below. This seems at odds with a lab that took great pains, to make many quantitative measurements. In aggregate these concerns are minor, and this manuscript is viewed suitable for publication once the comments below are addressed

1. On page 5, the authors note that they do not observe a pronounced ligand induced spectral change. Because TR-FRET is mentioned above, I would clarify that this is significant perturbation of the fluorine resonance in the ^{19}F NMR spectrum and not FRET.
2. On page 6, the authors note that the K502C protein yielded a well-functioning protein. This language is not specific enough. I would presume this is in regards to the activity assays. The authors tabulate ligand induced activities in the SI and so I would quantify the differences. For example, X-Y fold change in affinity of ligands. "Some reduction" in affinity is mentioned on line 127. But without a quantitative statement this is not helpful to the reader to evaluate how perturbing the modification is.
3. On line 119, the authors state there was preferential labeling to K502 over the native cysteine 313. However, this only appears when 2 eq. which can lead to incomplete labeling which the authors allude to in the experimental. It was unclear to what degree each protein was labeled with BTFA, which should be included if the fluorine is causing a perturbing effect. Protein-MS experiments could quickly identify this.
4. The authors derive K_i values from the Cheng-Prussoff equation to verify if ligand occupancy would still be greater than 97%. This equation can induce error in the calculated values and it is recommended that newer equations which can be found here: https://www.ncbi.nlm.nih.gov/books/NBK91992/#receptorbinding.Practical_Use_of_Fluores. The equations by Huang et al. and Kennakin et al. work well. Empirically, it may also help to simply titrate the protein sample with ligand to see when saturation occurs.
5. Minor: Figure 2 caption, figure 2s should be 2t in the legend.
6. On line 169. The D_2O solvent isotope effect studies were well done. This is known effect for the solvent accessibility of fluorine to the environment. Rather than saying a large effect, I would calibrate the readers for what are expected changes, for example in Evancics et al. 2007. Biochemical et Biophysica acta. chemical shifts as large as 0.1-0.25 ppm have been observed for highly solvent exposed resonances. In addition to the review in ref. 25, a primary literature example would be useful such as the one above. Because D_2O and H_2O solutions have an isotope effect on the pH. It was unclear if the authors controlled for this effect on the fluorine chemical shift.
7. The corresponding chemical shift changes are hard to observe in Figure 3 with the axes offset.

Stacking the spectra would be a better approach, similar to the SI or listing the $\Delta\delta$ above the resonance.

8. On page 11 the CEST experiments demonstrating slow chemical exchange are well done. The authors conclude that the bound population exists with two slowly interconverting protein conformations. These experiments appear to be done at saturating concentration. As a suggestion, a titration would be able to show if the ligands are binding one conformer preferentially, or if the exchange between the two populations is occurring equally in all cases.

9. On line 398, the authors note the weakening of Helix 12 to "lead to broad or distinct resonances." These lineshapes lead to two different conclusions for the timescale, it seems they mean broad.

10. It would be helpful if the authors include the chemical structures of the inhibitor molecules. In some cases they talk about covalent alkylations. It would be useful to see the electrophilic site on the molecule, or in other cases they mention a CF₃ group, but its not very helpful without seeing the molecule in question.

Reviewer #3 (Remarks to the Author):

Recommendation: This paper is publishable subject to major revisions noted.

Comments:

Christman et al investigated the PPARgamma LBD conformations using F19 NMR in junction with MD simulations. The F19 probe BTFA, attached Y505 part of H12, provided specific information about the H12 dynamics with multiple observed conformations (e.g., active, inactive, etc) in the absence/presence of various ligands. In parallel, long MD simulations show that various H12 conformations are populated on the landscape well. This is quite impressive because the large-scale motions were captured by brute-force simulations accurately. Here are some questions.

It is not clear how BTFA attaches to H12, relating to Fig. 2a. Which side of H12 is BTFA located, closer to ligand-binding pocket or the positive side? What is the evidence?

Both 19F-NMR and MD results alone are revealing. What is lacking is the direct comparison between them. Where do they agree? Where not? If not, can the NMR data be used to refine MD results?

Despite the significance of the topic itself, the manuscript is not among the well-written. It is full of grammar errors, etc. I would be happy to read it again after careful proof-reading.

We thank the reviewers for the time they spent reading the manuscript and providing very helpful suggestions about ways in which it could be improved. We have addressed each suggestion below. We have extensively modified the manuscript based on reviewer comments and based on Nature Communications author guidelines and manuscript checklist.

Reviewer #1 (Remarks to the Author):

The activity of nuclear receptors is regulated by ligands. Binding of ligands to the LBD of nuclear receptors affects the interactions between the AF-2 domain of the LBD (in helix 12) and transcriptional coregulators (i.e., coactivators and corepressors). There are various ligands (i.e., full and partial agonists, antagonists, and inverse agonists), coactivators and corepressors. Different ligands generate different transcriptional outcomes, although crystal structures revealed that the conformations of helix 12 are similar in LBD bound to different ligands. In an attempt to understand the structural mechanism responsible for the diverse activity of different ligands, the authors hypothesize that the helix 12 of LBD consists of a dynamic ensemble of conformations (that provide binding surface for coregulators) in response to different ligands. To test the hypothesis, the authors use PPAR γ LBD as a model, and ¹⁹F NMR, TR-FRET assay and molecular simulations as methodology. They utilized a short peptide containing one receptor-interacting motif from one coactivator (MED1) or one corepressor (NCoR), and a collection of 16 PPAR γ ligands to facilitate their studies. Based on the results from their studies, the authors claim that different ligands induce different coregulator binding surfaces: multiple thermodynamically accessible conformations (for partial agonists or no ligand) and one thermodynamically accessible conformations (for full agonists and inverse agonists). These data provide novel information helpful for understanding the different effects of different ligands, which are of interest to readers in the nuclear receptor research community, and the wider transcription field.

Addressing the following points will strengthen the author's conclusions:

We thank the reviewer for their suggestions and feel that addressing them has indeed strengthened our conclusions significantly.

1. The authors have utilized a short peptide from MED1 or NCoR for both the TR-FRET and NMR studies. At least one additional coactivator (e.g., PGC-1) and one additional corepressor (e.g., SMRT) should be used for representative ligand from each ligand type;

We chose to analyze representative ligands of each efficacy type with high affinity for PPAR γ (T0070907, MRL24, INT131, GW1929, rosiglitazone) and apo. We chose CBP as it has high affinity for PPAR γ and SMRT because it is another corepressor that is known to bind PPAR γ . We initially tried using TR-FRET to measure ligand efficacy in recruiting these coregulator peptides. This worked very well for CBP, but we saw only a slight increase in SMRT binding to wt PPAR γ upon titration with T0070907. We thus turned to fluorescence polarization (FP) to determine ligand effects on the binding affinity of SMRT for PPAR γ and found that this worked well; we also used FP to compare the SMRT, MED1 and NCoR binding affinities. We added these data to **Supplementary Figure 6 and 7** and **Figure 6**.

In addition, we determined the effect of CBP and SMRT binding on T0070907 and GW1929 bound PPAR γ and apo PPAR γ . These data were added to **Figure 7**. In completing this request, we noticed variability in the T0070907 spectrum using protein from different protein purifications. We therefore recollected all the T0070907 +/- coregulators spectra using the same protein preparation and included spectra from GW9662, which is a related ligand (**Figure 7**). We also performed a titration of T0070907 into PPAR γ^{K502C} -BTFA to assure that changes induced by T0070907 saturate (**Supplementary Figure 5**). We have included both the old and new T0070907 spectra overlaid in Supplementary Figure 10, and we point out the fact that we see this variability in the results where it reads:

"In separate protein preparations we did observe variation in T0070907 bound spectra (**Supplementary Fig. 10c**) this may be due to variable amounts of residual co-bound lipids as PPAR γ has a very large ligand binding pocket that can accommodate more than one bound ligand¹⁷."

Given the above reasoning, we replaced the T0070907 spectrum in **Figure 2** with the newer T0070907 spectrum as we believe this is the better spectrum. In addition, all of the correlations use the new T0070907 mean chemical shift.

2. In the TR-FRET assay (Figure 1), the basal activity (i.e. activity from solvent control; what is the solvent used to solubilize compound? DMSO?) should be included and clearly indicated (although the activity at the lower compound concentrations might be close to the basal activity);

We have included data demonstrating the ligand vehicle (DMSO) has a negligible effect on coregulator recruitment in the TR-FRET assay (**Supplementary Figure 5**) and reference these data in the figure legend of **Figure 1**. We have also added to the figure legend of **Figure 1** the information that DMSO is constant across the titration, which assures that there is no variable effect of the vehicle on the TR-FRET ratio displayed.

3. The authors noticed that the labeled proteins (PPAR γ K502C-BTFA and PPAR γ C313A,K502C-BTFA) have lower binding affinity to ligand and reduced ligand-induced recruitment of MED 1 and NCoR. While Pioglitazone is a full agonist for the wild-type PPAR γ (Figure 1, for MED1), it appears to function as a partial agonist for PPAR γ K502C-BTFA (Supplementary Figure 6a, for MED1). Please discuss how the slightly different ligand behavior between the wild-type/unlabeled and mutated/labeled PPAR γ might have affected the conclusion drawn from the NMR studies.

As the reviewer points out, we do see differences in relative ligand induced coregulator recruitment efficacy between wt and the mutants. We have run additional experiments with additional peptides, and have also included fluorescence polarization data to better understand the effect of labeling/mutations on the function of PPAR γ (these data also address reviewer 2's comment number 3). We present these data in **Supplementary Figure 6 and 7** and have added commentary in the results and discussion section about these data. We have added discussion (second to last paragraph of discussion) about how the functional differences observed between wt and mutant proteins would be expected to affect the NMR spectra we observe.

4. Figure 6 correlates data from the NMR to that from TR-FRET for the labeled PPAR γ -BTFA. Related to point #3 above, it might be useful to include the TR-FRET data for the wild-type PPAR γ (i.e., data from Figure 1) in the comparison.

We have included this comparison suggested by the reviewer in **Figure 6** and **Supplementary Figure 14**.

5. PPAR γ functions as a heterodimer with RXR. Please discuss whether and how the conclusions from this study might be affected if RXR is included in the assays?

We have included additional data (**Figure 8**) that gives an idea of what the effects of RXR α binding are. In general, RXR heterodimerization induces an increase in coactivator recruitment and decreases corepressor recruitment in TR-FRET. Small shifts are observed in the NMR spectra, except for T0070907 bound PPAR γ which has a larger upfield shift.

6. Page 25: the last sentence of the first paragraph appears to be incomplete.

We have moved this incomplete sentence about data availability to a section at the end of the methods entitled "Data availability."

7. Figure 1: does the color arrow indicate ligand potency? If so, the position of Troglitazone and Pioglitazone should be switched; similarly, for Supplementary Figure 6a, the list of ligands on the right can be better arranged to match the potency indicated by the curves (e.g., for MED1, Pioglitazone is listed on the top, but it is not the most potent ligand revealed by the curves).

The color in Figure 1 does not refer to potency, but instead relates to efficacy of coregulator recruitment. In the experimental system reported on here efficacy appears to be an indicator of ligand induced changes in affinity for a peptide. This can be inferred from the measurements of ligand induced changes in affinity we present here using fluorescence polarization and unpublished Isothermal titration calorimetry data. Relative coregulator binding efficacy is the main point we would like to convey in this figure, therefore we have left it unchanged. We have rearranged the order of the ligands in Supplementary figure 6a to be arranged by potency where possible (some ligands don't change baseline) as this will likely aid the reader as the reviewer points out. We have also added additional supplementary tables that show the EC50 values for TR-FRET (**Supplementary Tables 3 and 4**).

Reviewer #2 (Remarks to the Author):

The manuscript by Chrisman et al. is a novel application of fluorine NMR spectroscopy for deciphering the conformational landscape of a nuclear receptor. The authors use 16 pharmacological inhibitors as well as native coactivator and repressor ligands to provide a rich data set to assess the different modes of activation. Due to the highly responsive fluorine nucleus, ¹⁹F NMR is becoming increasingly popular for addressing difficult questions in structural biology. The current application to a nuclear receptor is of significant biomedical importance, and the authors exploited from a simple resonance, both the chemical shift and D₂O dependence, and chemical exchange

experiments by fluorine NMR combined with molecular dynamics and FRET experiments to build a compelling picture of PPAR γ conformational dynamics. The conclusions in general seem sound. Although the experiments were well done, there were sections where clarity could be improved. For example, there are a variety of

qualitative statements made when quantitative would be more appropriate. Uses of the phrases “well-functioning”, “similar patterns”, “lower potency”, “some reduction of affinity” “around 1/s” could be strengthened if actual values were discussed rather than burying the values in a supporting information table. In some cases this seems to be smoothing over potential perturbing effects from fluorine labeling. For example in Figure S6 the correlation plots don't seem to be one-to-one, so as stated it is hard to interpret what similar patterns would be that give rise to similar function. Other minor concerns are listed below. This seems at odds with a lab that took great pains, to make many quantitative measurements. In aggregate these concerns are minor, and this manuscript is viewed suitable for publication once the comments below are addressed

We thank the reviewer for the many insightful comments and suggestions as responding to them has strengthened this paper considerably.

Regarding the reviewer's comment “For example in Figure S6 the correlation plots don't seem to be one-to-one, so as stated it is hard to interpret what similar patterns would be that give rise to similar function.”

There are two types of differences between mutants and wt as detected by TR-FRET.

The first is that efficacy values for NCOR recruitment to PPAR γ ^{C313A,K502C}-BTFA have low correlation with wt PPAR γ -LBD values ($R^2=0.11$), in other words the labeling/mutations have variable impact on NCOR binding for PPAR γ ^{C313A,K502C}-BTFA that depends on the ligand. In contrast, PPAR γ ^{K502C}-BTFA recruitment of NCOR correlates well with PPAR γ ($R^2=0.8$). Furthermore, PPAR γ MED1 and CBP TR-FRET values correlate very well with both PPAR γ ^{K502C}-BTFA and PPAR γ ^{C313A,K502C}-BTFA (**Supplementary Figure 6**). For this reason, we rely on data from PPAR γ ^{K502C}-BTFA to correlate NMR detected structure with function. We utilize PPAR γ ^{C313A,K502C}-BTFA to confirm specific labeling of PPAR γ ^{K502C}-BTFA and in cases where increased signal to noise is required as in the chemical exchange experiments. We have modified **Supplementary Figure 6** to make the correlation between wt and the labeled mutants more legible.

The second is the difference in absolute TR-FRET ratios between PPAR γ ^{K502C}-BTFA and wt. Absolute TR-FRET ratios are sensitive to protein concentration, which are not exact between necessarily different protein preparations. In addition, the PPAR γ ^{K502C}-BTFA is not fully labeled (64%) and the non-BTFA labeled PPAR γ ^{K502C} does not contribute meaningfully to the TR-FRET signal which will affect the observed maximum TR-FRET ratio. This limits the signal to noise of the assay but does not indicate that PPAR γ ^{K502C}-BTFA has decreased or changed function in binding coregulators or ligands. Of note is that this non-BTFA labeled protein is functional as measured by FP (**Supplementary Figure 4 and 7**; see response to comment 3 below). These data are now referenced in the results.

1. On page 5, the authors note that they do not observe a pronounced ligand induced spectral change. Because TR-FRET is mentioned above, I would clarify that this is significant perturbation of the fluorine resonance in the 19F NMR spectrum and not FRET.

We have changed the sentence to read “Q498C caused protein instability, whereas BTFA attached to Y505C did not show pronounced ligand induced changes to the ¹⁹F NMR spectra of the PPAR γ LBD, presumably due to its position at the unstructured C-terminus of helix 12 (**Supplementary Figure 1**).”

2. On page 6, the authors note that the K502C protein yielded a well-functioning protein. This language is not specific enough. I would presume this is in regards to the activity assays. The authors tabulate ligand induced activities in the SI and so I would quantify the differences. For example, X-Y fold change in affinity of ligands. “Some reduction” in affinity is mentioned on line 127. But without a quantitative statement this is not helpful to the reader to evaluate how perturbing the modification is.

We have made extensive changes to the paragraph that describes the effects of the mutations and labeling on ligand binding and coregulator recruitment including adding numbers in place of qualitative words to describe similarities and differences to wt PPAR γ LBD. In addition, we have collected additional data to better quantify what the effects of labeling are. (**Figure 6, Supplementary Figure 6 and 7**).

3. On line 119, the authors state there was preferential labeling to K502 over the native cysteine 313. However, this only appears when 2 eq. which can lead to incomplete labeling which the authors allude to in the experimental. It was unclear to what degree each protein was labeled with BTFA, which should be included if the fluorine is causing a perturbing effect. Protein-MS experiments could quickly identify this.

We have measured the percent labeling of the PPAR γ ^{K502C} by BTFA that we used in the TR-FRET assays (3x BTFA). One of our ligands (MRL24) contains a CF₃ group, just as BTFA does. By integrating the signal from the BTFA and that of bound MRL24 we found that 64% of this protein is labeled with BTFA at K502 (**Supplementary Figure 4**).

We also ran experiments to determine what effect, if any, the BTFA label itself has on coregulator recruitment (TR-FRET) and affinity (Fluorescence polarization; FP). For unknown reasons PPAR γ ^{K502C} without the BTFA does not show any response in TR-FRET to any ligands, but this same protein behaves fine in FP and when labeled with BTFA in TR-FRET (**Supplementary Figure 4**). Despite using 5mM TCEP, this may be a result of disulfide linkage with cysteines in the antibodies or streptavidin that are included in TR-FRET but not FP, or some other unknown problem. We thus turned to FP to assess the effect of the BTFA label on function. As measured by FP the BTFA label has very little, if any effect on apo or drug induced affinity for coregulators. These data are now included in **Supplementary Figure 4** and **Supplementary Figure 7**. These data are now referenced in the results.

4. The authors derive K_i values from the Cheng-Prusoff equation to verify if ligand occupancy would still be greater than 97%. This equation can induce error in the calculated values and it is recommended that newer equations which can be found here: https://www.ncbi.nlm.nih.gov/books/NBK91992/#receptorbinding.Practical_Use_of_Fluores. The equations by Huang et al. and Kennakin et al. work well. Empirically, it may also help to simply titrate the protein sample with ligand to see when saturation occurs.

As we do all of our NMR remotely in New York, ligand titrations are difficult for us to perform and so we did not do this. We thank the reviewer for the suggested equations to more accurately quantify K_i. We have used the Kenakin equation (Kenakin, TP 1993 in *Pharmacologic analysis of drug/receptor interaction*, 2nd ed., New York:Raven p. 483.)

Of note the reference https://www.ncbi.nlm.nih.gov/books/NBK91992/#receptorbinding.Practical_Use_of_Fluores appears to have an extra negative sign on the bottom part of the equation; (Lo)(Ro) + Lb(-Ro-Lo+Lb-Kd). Therefore, we used the following corrected equation without this negative sign to calculate K_i.

$$K_i = \frac{(Lb)(IC_{50})(Kd)}{(Lo)(Ro) + Lb(Ro - Lo + Lb - Kd)}$$

5. Minor: Figure 2 caption, figure 2s should be 2t in the legend.

This has been corrected.

6. On line 169. The D₂O solvent isotope effect studies were well done. This is known effect for the solvent accessibility of fluorine to the environment. Rather than saying a large effect, I would calibrate the readers for what are expected changes, for example in Evancics et al. 2007. *Biochemical et Biophysica acta*. chemical shifts as large as 0.1-0.25 ppm have been observed for highly solvent exposed resonances. In addition to the review in ref. 25, a primary literature example would be useful such as the one above. Because D₂O and H₂O solutions have an isotope effect on the pH. It was unclear if the authors controlled for this effect on the fluorine chemical shift.

We thank the reviewer for pointing this out as we failed to correct for the isotope effect on pH in our original data. We repeated the D₂O experiment with pD/pH values corrected for isotope effects (using the method suggested in P. Glasoe and F. Long *The Journal of Physical Chemistry* 1960 64:188-190). We found that most peaks moved similarly, however there were some differences between the uncorrected and corrected experiments.

We have also included the mentioned reference and another similar reference and changed the text appropriately.

7. The corresponding chemical shift changes are hard to observe in Figure 3 with the axes offset. Stacking the spectra would be a better approach, similar to the SI or listing the $\Delta\delta$ above the resonance.

We have made these changes.

8. On page 11 the CEST experiments demonstrating slow chemical exchange are well done. The authors conclude that the bound population exists with two slowly interconverting protein conformations. These experiments appear to

be done at saturating concentration. As a suggestion, a titration would be able to show if the ligands are binding one conformer preferentially, or if the exchange between the two populations is occurring equally in all cases.

The downfield (left shifted) peaks characteristic of PPAR γ bound to ciglitazone, pioglitazone and troglitazone (which possess two well separated peaks) overlay the small upfield apo peak which complicates their use in this experiment. We thank the reviewer for this suggestion as it spurred our interest in this type of experiment.

[Redacted]

We appreciate the reviewer's suggestion and we look forward to using this approach in future work.

9. On line 398, the authors note the weakening of Helix 12 to "lead to broad or distinct resonances." These lineshapes lead to two different conclusions for the timescale, it seems they mean broad.

We have corrected this in the manuscript (by removing "or distinct")

10. It would be helpful if the authors include the chemical structures of the inhibitor molecules. In some cases they talk about covalent alkylations. It would be useful to see the electrophilic site on the molecule, or in other cases they mention a CF₃ group, but it's not very helpful without seeing the molecule in question.

This is now included in **Supplementary Figure 16**.

Reviewer #3 (Remarks to the Author):

Recommendation: This paper is publishable subject to major revisions noted.

Comments:

Christman et al investigated the PPAR γ LBD conformations using F19 NMR in conjunction with MD simulations. The F19 probe BTFA, attached to the Y505 part of H12, provided specific information about the H12 dynamics with multiple observed conformations (e.g., active, inactive, etc) in the absence/presence of various ligands. In parallel, long MD simulations show that various H12 conformations are populated on the landscape well. This is quite impressive because the large-scale motions were captured by brute-force simulations accurately. Here are some questions.

It is not clear how BTFA attaches to H12, relating to Fig. 2a. Which side of H12 is BTFA located, closer to ligand-binding pocket or the positive side? What is the evidence?

We thank the reviewer for pointing out the lack of clarity for this key point. We have redone supplementary Figure 3 to better show the dominant orientations of the BTFA CF₃ group during the ~12 μ s simulation of PPAR γ ^{K502C}-BTFA bound to the agonist GW1929.

Both 19F-NMR and MD results alone are revealing. What is lacking is the direct comparison between them. Where do they agree? Where not? If not, can the NMR data be used to refine MD results?

We have modified figure 9 (which was figure 8 in the previous version of the manuscript) to clarify the similarities and differences between the NMR and the MD results. We have also added additional commentary in the results and discussion regarding comparison of the NMR and MD results. In this manuscript we are limited to qualitative comparison of the NMR and MD data, which consists of comparing the width and number of 19F NMR peaks with the outcomes of multiple independent simulations

Unfortunately, at this point we are unable to restrain MD runs based on NMR values as we do not have quantitative information such as distance between atoms. We are working on obtaining distance restraints that could be used for this purpose and plan on including these in a future publication. In addition, we are working on improving the accuracy of ¹⁹F NMR chemical shift prediction from MD (of BTFA labeled PPAR γ) in collaboration with David Case

(Rutgers) using a variant of his chemical shift prediction algorithm (SHIFTS). However, we cannot as of yet accurately predict ^{19}F NMR shifts from simulations.

We point these ideas out in the discussion: *“Overall it is encouraging that these non-converged simulations qualitatively agree with ^{19}F NMR and provide a glimpse of possible conformations that comprise a portion of the observed ^{19}F NMR spectra, however quantitative comparison between the ^{19}F NMR spectra and converged simulations remains a future challenge.”*

Despite the significance of the topic itself, the manuscript is not among the well-written. It is full of grammar errors, etc. I would be happy to read it again after careful proof-reading.

We thank the reviewer for pointing out grammatical errors and the difficulty of reading. We have done extensive rewriting to improve the readability and to fix grammatical errors. We hope that this has improved the quality of the manuscript.

REVIEWERS' COMMENTS:

Reviewer #1 (Remarks to the Author):

The authors have satisfactorily addressed all my comments by either providing additional experimental data or discussions.

Reviewer #2 (Remarks to the Author):

I have reassessed the revised manuscript and am satisfied with the authors efforts both explanatory and experimentally to address my concerns from the prior submission. The more quantitative evaluations have improved the clarity of their analysis regarding functional perturbations. As a minor point, the added molecular structures in the supporting information could be improved regarding consistent formatting and correct bond angles (e.g. molecule d).

Second response to reviewers

REVIEWERS' COMMENTS:

Reviewer #1 (Remarks to the Author):

The authors have satisfactorily addressed all my comments by either providing additional experimental data or discussions.

We thank the reviewer for the time and effort they took to suggest improvements.

Reviewer #2 (Remarks to the Author):

I have reassessed the revised manuscript and am satisfied with the authors efforts both explanatory and experimentally to address my concerns from the prior submission. The more quantitative evaluations have improved the clarity of their analysis regarding functional perturbations. As a minor point, the added molecular structures in the supporting information could be improved regarding consistent formatting and correct bond angles (e.g. molecule d).

We thank the reviewer for the time and effort they took to suggest improvements. We have utilized the chemdraw template suggested by Nature Communications to reformat the structures, we have used bond angles as suggested by chemdraw and have arranged the conformation to be similar to previously published work. We have also made the rings (if present) in the structures the same relative size so that the presented sizes more accurately reflect the actual relative molecular sizes.